# Soft iEP: On the Exploration Inefficacy of Gradient-Based Strong Lottery Exploration

## Abstract

Edge-popup (EP) is a de facto algorithm to find *strong lottery tickets (SLT)*, the sparse subnetworks that achieve high performance *without weight updates*. EP finds the subnetworks by optimizing a score vector representing the importance of each edge, and selects subnetworks given optimized scores. This paper first shows that such a simple gradient-based method results in a suboptimal solution due to the existence of *dying edges*. Specifically, we show that most edges are *never* selected during the search process, i.e., EP might be trapped around the local minima nearby random subnetworks and need help to search the entire spaces of subnetworks effectively. We then propose a *soft iterative edge-pop (Soft iEP)* as a simple mechanism to better explore search spaces. Unlike the standard iterative pruning that masks out a certain amount of edges and thus induces a similar problem to the dying edges, Soft iEP *do not* disable the bottom edges at each cycle, i.e., leave a chance to be selected at the end regardless of whether it was chosen at the former cycle. Empirical validations show that iEP with soft pruning stably outperforms both EP and iEP w/ hard pruning on ImageNet, CIFAR-10, and CIFAR-100 and reduces dying edges. Our results also provide new insight into why iterative pruning helps to find good sparse networks.

## 1 Introduction

Ramanujan et al. (2020); Chijiwa et al. (2021); Yeo et al. (2023) demonstrated that a randomly initialized neural network contains sparse subnetworks that achieve comparable performance with fully-trained dense networks. For example, Ramanujan et al. (2020) shows that randomly initialized WideResNet50 contains subnetworks that achieve comparable performance to trained ResNet34 without updating weights. Such sparse networks are called as *strong lottery tickets (SLT)*, since it shares the core concept with *lottery tickets hypothesis* (Frankle and Carbin, 2019) in a sense that both suggest a randomly initialized over-parameterized networks contain subnetworks that attain good properties. It has been raising attention on the phenomenon of SLT, not only its intriguing property of the overparameterized random networks (Du et al., 2018; Allen-Zhu et al., 2019), but also several practical benefits including that (1) SLT is robustness to the binarization or quantization and weights to save the memory footprint (Diffenderfer et al., 2021; Diffenderfer and Kailkhura, 2021), and (2) unlearned weight can be reconstructed only by storing random seed and binary masks (Okoshi et al., 2022). Hirose et al. (2022) proposed specialized hardware to accelerate inference using the property.

Many subsequent works support the existence of SLT, both empirically (Ramanujan et al., 2020; Chijiwa et al., 2021; Sreenivasan et al., 2022; Yeo et al., 2023) and theoretically (Malach et al., 2020; Pensia et al., 2020; Orseau et al., 2020; Burkholz, 2022; da Cunha et al., 2022). Although the problem of discovering a good subnetwork can be naturally cast as a combinatorial optimization (Korte et al., 2011), it is usually intractable given a huge number of parameters. Instead, most prior methods rely on *edge-pop (EP)* algorithm (Ramanujan et al., 2020), which cast the structure search as a sort of stochastic optimization. Specifically, EP first assigns a real-valued score for each edge and updates its score via the loss calculated using subnetworks selected based on the scores (e.g., by a threshold). Prior studies show that EP can find SLT that is comparable with fully-trained networks on various model sizes (from small conv to wide ResNet), datasets (including ImageNet), and tasks (classification and generation (Ramanujan et al., 2020; Yeo et al., 2023)).

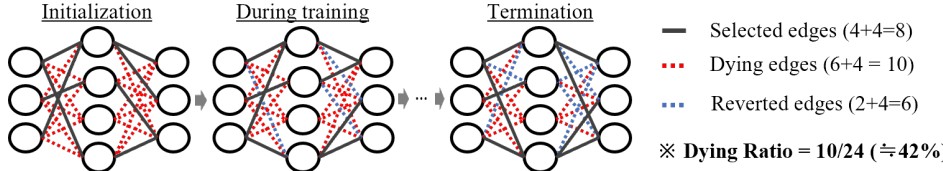

Figure 1: Visualization of the dying edges. Black edges are selected at the current iteration, blue edges are currently masked but selected at least once, and red edges (dying edges) are masked and never selected at the time. The ratio of dying edges (dying ratio) is 42% in the example.

Despite the dominant use of EP, such a simple gradient-based method is not designed for combinatorial optimization and leads to suboptimal subnetworks. To illustrate the problem, we introduce the notion of *dying edge*, which represents an edge that is *never* activated during the optimization process of EP (red edges in Figure 1). Our empirical results show that the standard EP induces many dying edges, e.g., *40%* of edges are dying when training ResNet18 on CIFAR-10 when pruning approximately half the edges, which leads to suboptimal performance. In other words, EP only searches areas nearby randomly selected initial subnetworks, and can not fully explore entire search spaces.

Based on the observation, we propose *soft iterative edge-popup (Soft iEP)*, which couple the success in the standard lottery tickets hypothesis (LTH) and the dying edge problem introduced in this paper. Similar to the iterative pruning in the standard LTH, Soft iEP repeatedly applies the edge-pop algorithm with a gradually increasing prune rate after each cycle. However, the standard convention in LTH, which we referred to as *hard pruning* that masks out a certain amount of edges before starting the next cycle based on the current pruning ratio $p$, does not necessarily or even *hurts* the performance in SLT. This is because hard pruning might filter out edges at the early cycle of iterative pruning and induce a similar problem to the dying edge problem. Instead, we propose a soft version of iterative pruning, called *soft pruning*, which gradually increases the pruning rate but does not entirely disable the bottom $p$ edges at each cycle. In other words, all edges can be selected at the end, regardless of whether it was chosen at former cycles.

Note that it is true that the iterative pruning is well-known in the standard lottery tickets hypothesis (LTH) (Frankle and Carbin, 2019; Paul et al., 2022); however, its benefits have not been fully investigated in the context of SLT. Indeed, our results show that the standard iterative pruning used in the LTH does not work well on the SLTH setup, especially in high sparsity region. Moreover, the reason why iterative pruning helps to find LT is not fully understood (Zhang et al., 2021; Paul et al., 2022), making it nontrivial whether the same approach works well on different setups. Our results suggest that understudied benefits of iterative pruning to ease the exploration inefficiency of the gradient-based optimization.

Our empirical validations on image classification tasks using various datasets (ImageNet (Deng et al., 2009), CIFAR-10, and CIFAR-100 (Krizhevsky et al., 2009)) show that iEP with soft pruning stably outperforms both EP and iEP w/ hard pruning on all datasets and model size (ResNet18, ResNet50, ResNet101, and WideResNet50). Notably, it discovered a subnetwork that is sparser than ResNet-34 but exceeds the performance of trained dense ResNet34 by over 2.4% in the accuracy of ImageNet (76.0% with 20M parameters). We also investigate the benefit of softness in iterative pruning from the perspective of the exploration efficiency.

## 2 BACKGROUND

**Definition 2.1 (Strong Lottery Ticket Hypothesis (SLTH)).** Let $g(\boldsymbol{x}; \boldsymbol{\theta})$ be a target dense network, e.g. trained via standard gradient descent, with a test accuracy $a$. Consider a dense network $f(\boldsymbol{x}; \boldsymbol{\theta}_0)$ whose weights are randomly initialized ($\boldsymbol{\theta}_0 \sim \mathcal{D}_\theta$). There exists a mask $\boldsymbol{m} \in \{0, 1\}^{\|\boldsymbol{\theta}_0\|}$ such that $f(\boldsymbol{x}; \boldsymbol{m} \odot \boldsymbol{\theta}_0)$ reaches a test accuracy $a'$ without any training, where $a' \geq a$, and $\|\boldsymbol{m}\|_0 \ll |\boldsymbol{\theta}|$.

Informally, strong lottery tickets hypothesis (SLTH) (Zhou et al., 2019; Ramanujan et al., 2020; Malach et al., 2020) states that the randomly initialized networks themselves contain sparse subnetworks that achieve comparable performance with trained networks. This paper refers to the mask itself or the sparse subnetworks $f(\boldsymbol{x}; \boldsymbol{m} \odot \boldsymbol{\theta}_0)$ as *strong lottery tickets (SLT)*.

**Difference between SLTH and LTH.** As the name suggests, the SLTH is closely related to the lottery tickets hypothesis (LTH) (Frankle and Carbin, 2019; Paul et al., 2022). Both hypotheses suggest the existence of "good" and sparse subnetworks in sufficiently large, randomly connected

neural networks but differ in how they measure the "goodness" of subnetworks. Namely, The LTH considers the subnetworks that reach good performance *after training*. SLTH considers the stronger condition, where the subnetworks achieve good performance *without training*. Due to the difference, they employ different methods to find subnetworks. Namely, LT is discovered based on the magnitude of the weight after training, and SLT is discovered based on the score value that represents the importance of each weight. This paper focuses on the finding of SLT.

**Iterative Pruning and Rewinding.** In LT, *iterative pruning* and *rewinding* are two components dominantly used in practice to improve the performance of the subnetworks. To be more specific, most studies in LT use *iterative magnitude pruning*, which iteratively prune weights based on the magnitude of weights after training the network for a certain number of epochs. At the end of each cycle, the weights and/or learning rate is *rewinded* to the certain epochs of training, and repeat the process $K$ times, which is known to be vital. However, it is not tailored for SLT, and the effectiveness of these concepts in SLT is not fully investigated. Besides, the reason why iterative pruning is necessary to find good subnetworks is still not fully understood (Zhang et al., 2021; Paul et al., 2022). We discuss the benefit of iterative pruning with the dying edge problem introduced in this paper.

**Theoretical Finding about SLTH** Compared to LTH, SLTH is still under-investigated; however, the existence of SLT has been validated both theoretically and empirically. Malach et al. (2020) provide a first proof of SLTH for fully connected networks with ReLU activation functions. They show that a subnetwork hidden in a sufficiently large randomly initialized neural network can approximate any target network with a high probability. Subsequently, Pensia et al. (2020); Orseau et al. (2020) derive tighter bounds regarding the required network size from approximating a target network. More recently, Burkholz et al. (2021) proved the existence of a much stronger version of lottery tickets, called *universal tickets*, which are subnetworks that can be reused across a variety of tasks without further training on the target tasks. However, these studies focus on the existence of strong lottery tickets and do not provide an analysis of the behavior of EP widely used in empirical studies.

**Empirical Finding about SLTH** Besides the theoretical analysis, several studies have empirically shown the existence of SLT. While it has not yet been called strong lottery tickets, a study by (Zhou et al., 2019) is a pioneer work that showed it is possible to train networks only via weight masking. However, due to the lack of a sophisticated masking algorithm, its performance was not good as with the dense training, and empirical experiments are limited to small-scale datasets. Later, Ramanujan et al. (2020) proposed EP and showed that EP efficiently discovered SLT that performed on par with the dense training. More recently, Yeo et al. (2023) investigates strong lottery tickets in generative models, specifically generative adversarial networks, and shows that EP with moment matching loss (Gretton et al., 2006) can discover subnetworks that can generate good images on CelebA, FFHQ, and LSUN with the similar amount of final parameters.

Ramanujan et al. (2020) proposed an algorithm called **Edge-Popup (EP)**, and subsequent works also use EP. In the EP, a popup score $s$ ($s \in \mathbb{R},\ s \geq 0$) is assigned to each weight in the network, representing the contribution of the weight to the network. The score is usually drawn from a uniform distribution. Given a pruning ratio $k$, each layer's top-$(1-k)$ highest scores are selected in the forward path. The pruning ratio $k$ is usually fixed during training and is the same for each layer, but it can be dynamic and different for each layer. The prediction is then used to calculate the loss. Let $s$ as a randomly initialized scores corresponding to initialized weights $\boldsymbol{\theta}_0$, $\boldsymbol{m}_k^s \in \{0,1\}^{|\boldsymbol{\theta}_0|}$ as a mask vector deterministically decided by $s$ and prune rate $k$. While initial weights $\boldsymbol{\theta}_0$ can be sampled from arbitral distributions, we mainly sample it from a signed constant distribution $\mathcal{U}$ introduced by Ramanujan et al. (2020). Specifically, we set each weight as a constant except for randomly chosen sign (+ or -), which means that the weights of layer $l$ can be represented like $\boldsymbol{\theta}_0^l = [\sigma_0^l, -1 * \sigma_1^l, -1 * \sigma_1^l \cdots, 1 * \sigma_{n_l-1}^l] = \sigma^l * [1, -1, -1, \cdots, 1]$. Following (Ramanujan et al., 2020), we use the standard deviation of Kaiming Normal (He et al., 2015) as the constant value.

Given a batch of data $\{(\boldsymbol{x}_i, y_i)\}_{i=1}^{B}$, where $B$ is a batch size, EP updates $s$ by

$$\boldsymbol{s}^* = \arg\min_{\boldsymbol{s}} \sum_i^B L(f(\boldsymbol{x}_i; \boldsymbol{m}_k^s \odot \boldsymbol{\theta}_0), y_i), \tag{1}$$

where $\odot$ represents element-wise product and $L$ represents a loss function. EP uses stochastic gradient descent (SGD) to solve the above optimization. As the gradient can not be naively computed for unselected weights, EP uses a straight-through estimator (Bengio et al., 2013) to calculate the gradient for scores; i.e., the gradient of the sign function is treated as the identity.

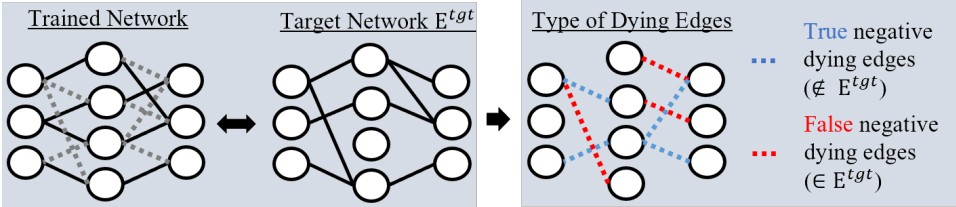

Figure 2: Illustration of *true negative* dying edges (blue) and *false negative* dying edges (red). The former is a dying edge that does not contribute to the prediction, i.e., not included in the target network. The latter is a dying edge included in the target network and should be selected.

## 3 EXPLORATION INEFFICIENCY OF EP

As mentioned in the introduction, the dying edge is referred to as the edge that is never selected until the termination of EP. Figure 1 visualize the concept of dying edge. In Figure 1, the black edges represent *selected edges*, whose score $s$ exceeds a threshold and thus is used for the current forward path. The *selected edges* can be changed during the optimization, as the corresponding score matrix $s$ changed during optimization. We referred to such edges as *reverted edges* (blue dashed line). Finally, at the termination, some edges might never be selected at all iterations. We call such edges as *dying edges* since they never contributed to the forward path. Since EP uses a straight-through estimator, scores corresponding to masked edges are updated, while the gradients are coarse compared to the actual gradients. More generally, at each iteration $t$, we can calculate the dying ratio $d_t$ given the sum of the history of mask $\boldsymbol{m}_{\leq t} = \sum_{i \in 1 \cdots t} \boldsymbol{m}_i$:

$$d_t = \frac{\sum_{x \in \boldsymbol{m}^t} \delta_{x,0}}{|\boldsymbol{\theta}_0|}, \tag{2}$$

where $\delta_{x,0}$ is a Kronecker delta. By definition, $d_t$ is monotonically decreased, and the initial dying ratio $d_0$ is always equal to the pruning rate $k$, i.e., $d_t \leq d_0 = k$. As a corollary, $k - d_t$ represents the percentage of edges that are first masked but selected at least once during the training. The latter metric is handy for comparing the behavior among different pruning ratios.

Intuitively, a higher dying ratio is a bad sign and indicates the exploration inefficacy of an algorithm, as it means that many edges need to be better tested during the entire search process. However, it is possible that some edges are actually useless and should be filtered out early as possible. In other words, there are two types of dying edges: (1) *true negative dying edge* that is not included in the target network and useless, and (2) *false negative dying edge* that is included in the target network and should not be masked. Figure 2 visualize the distinction between true negative and false negative dying edges. Due to two types of dying edges, a low dying ratio could also lead to performance degradation due to exploitation inefficacy. For example, if we choose the edges completely random at every iteration, the dying ratio must be zero at some iteration, but it will not give good performance. Unfortunately, it is impossible to distinguish these two types in practice because the target network is unknown. Besides, even if we know or can assume the target network, an algorithm can find equally good but different networks due to the symmetry of neural networks. Therefore, we mainly focus on the simple dying ratio Equation 2 and investigate its relationship with the final performance.

**Case study: EP induces a surprisingly high dying ratio.** To analyze the behavior of EP, we first quantify the dying ratio when applying EP on CIFAR-10 and SVHN. Specifically, we applied EP (Equation 1) to ResNet-18 following the configurations of (Ramanujan et al., 2020). Namely, we used SGD optimizer for 100 epochs with batch size 128, weight decay 0.0001, momentum 0.9, and learning rate 0.1, decayed using a cosine annealing schedule. Table 1 show the final dying ratio $d_T$ and $k - d_T$ with various pruning ratios $k$ (20% to 83.2%). We repeated the same experiment with different seeds four times and showed the average values. Since the standard deviation is extremely low (e.g., 1e-4 when $k = 0.2$), we omit it for simplicity. The table shows that $d_T$ naturally increases as we use larger $k$. For example, when $k = 0.200$, the dying ratio is 0.028 for CIFAR-10, but when $k = 0.488$, the dying ratio is 0.293, which means almost half of the masked edges are never selected. In addition, Table 1 show that $k - d_T$, which represents the fraction of edges that are masked at the beginning of training but selected at the end, is around 0.20 in CIFAR-10 regardless of the choice of

Table 1: Dying ratio $d_T$ when applying EP with different pruning ratio $k$. $k - d_T$ represents the percentage of edges that are first masked but selected at least once during the training.

| $k = d_0$ | .200 | .360 | .488 | .590 | .672 | .738 | .790 | .832 |
|---|---|---|---|---|---|---|---|---|
| $d_T$ | .028 | .166 | .293 | .395 | .479 | .540 | .606 | .656 |
| $k - d_T$ | .172 | .194 | .195 | .195 | .193 | .190 | .184 | .176 |

Table 2: Effect of hyperparameters (weight and score initialization) on test accuracy and dying ratio (DR). As default parameters, we use (score initialization, weight initialization) = (Kaiming Uniform (uniform), Signed Kaiming Constant (sc)), and the pruning rate is set at 0.7 in all experiments.

| | | Conv8 | | | | ResNet18 | | | |
|---|---|---|---|---|---|---|---|---|---|
| | | CIFAR-10 | | SVHN | | CIFAR-10 | | SVHN | |
| $s_0$ | $\theta_0$ | Test Acc | DR | Test Acc | DR | Test Acc | DR | Test Acc | DR |
| scaled sc | uniform | **89.61** | **41.77** | **96.16** | **34.07** | 93.19 | **52.31** | 96.77 | **50.64** |
| scaled sc | normal | 87.45 | 50.11 | 95.76 | 45.47 | 92.62 | 59.18 | 96.42 | 57.98 |
| sc | uniform | 88.02 | 42.01 | 95.92 | 35.70 | **93.22** | 52.85 | **96.78** | 51.37 |
| normal | uniform | 87.52 | 46.38 | 95.63 | 40.81 | 92.43 | 54.57 | 96.73 | 52.69 |

$k$. Since the subnetworks are first selected entirely random at the first iteration, the results suggest that EP might be trapped around the local minima nearby random subnetworks and need help to search the entire spaces of subnetworks effectively. We also tested how the initialization of $s$ and $\theta$ affects the dying ratio in Table 2. While the choice of initialization affects the dying ratio, the default parameter is the best or on par in almost all cases (highest accuracy and lowest dying ratio). More discussion on how the choice of the general hyperparameter (including learning rate, batch size, and weight decay) could affect the dying ratio is discussed in the subsection A.2.

## 4 PROPOSAL: SOFT ITERATIVE EDGE POPUP

In this section, we propose a simple remedy to the dying edge problem induced by EP. Inspired by the success of iterative pruning in the standard LTH, we propose *soft iterative edge-pop* to alleviate the dying edge problem and facilitate better exploration. We first introduce iEP with *hard pruning*, which follows the standard convention in LTH. We then propose a new mechanism called *soft pruning* to solve the problem of hard pruning from the perspective of the dying edges.

### 4.1 ITERATIVE EDGE POPUP (IEP) WITH HARD PRUNING

Suppose we want to prune the edges of a network $\theta_0$ by a fraction of $k$. As described in section 2, the standard edge-pop always prunes edges by the fraction of $k$ from the beginning to the end, but $k$ can be dynamically changed during the search process. Iterative edge popup first masks $p << k$ percent of weights from the entire search space $\theta_0$, and then at the second cycle, i.e., generate $m_1$ using the standard EP with the pruning rate $p$. At the second cycle, iEP again masks $p$ percent of weights from $m_1 \odot \theta_0$, and repeat the process until we reach the desired pruning ratio. We refer to $p$ as *shrinkage rate* to distinguish it from the target prune rate $k$. With the initial pruning ratio $m$, $(1 - m)(1 - p)^{i-1}$ weights are remaining after $i$-th cycle. By default, we set $p = m$ for simplicity.

While the general concept of iterative pruning is simple, there are several ways to repeat the process.

**Fine tuning (FT)** Fine-tuning starts the next cycle with the scores trained in the prior cycle. For the learning rate $\gamma$, we use the final learning rate in the first iteration for the subsequent iterations, following the convention in LTH (Li et al., 2017; Liu et al., 2019). Note that, in the case of the LTH, it is shown that fine-tuning-based iterative pruning can cause significant degradation (Renda et al., 2020). Instead, most studies in LTH use a technique called *rewinding*, which restarts the optimization from the initial weights $\theta_0$, not the trained weights $\theta_T$. Since we do not train weights, the same approach is invalid, but we tested the following three rewinding approaches.

**Score rewinding (SRw)** Score rewinding retrains by rewinding both the learning rate and the scores. The initial scores $s_0$ are used as the initial value of the scores for every $i$-th cycle, and the learning rate and scheduler are the same as those used in the first iteration. In the lottery hypothesis,

Table 3: Comparison of different iterative pruning algorithms tested in the paper. At $i$-th cycle, hard iEP inheret $\boldsymbol{m}^*_{i-1}$ and select $p$ percent of edges from $\boldsymbol{m}^*_{i-1} \odot \boldsymbol{w}_0$. Contrary, soft iEP uses $\boldsymbol{s}^*_{i-1}$ as the initialization of score vectors but does not limit the search spaces. $\boldsymbol{s}_0$: initial scores, $\boldsymbol{s}^*_{i-1}$: learned scores at prior cycle, $\gamma$: initial learning rate, and $\gamma_{i-1}$: learning rate at the end of cycle $(i-1)$.

| Algorithm | pruning rate $k_i$ | score init $\boldsymbol{s}_{k_i}$ | learning rate | search space |
|---|---|---|---|---|
| EP | $1-(1-p)^i$ | $\boldsymbol{s}_0$ | $\gamma$ | $\boldsymbol{\theta}_0$ |
| Hard iEP w/ FT | $p$ | $\boldsymbol{s}^*_{i-1}$ | $\gamma^*_{i-1}$ | $\boldsymbol{m}^*_{i-1} \odot \boldsymbol{\theta}_0$ |
| w/ SRw | $p$ | $\boldsymbol{s}_0$ | $\gamma$ | $\boldsymbol{m}^*_{i-1} \odot \boldsymbol{\theta}_0$ |
| w/ SRi | $p$ | $\tilde{\boldsymbol{s}}_0$ | $\gamma$ | $\boldsymbol{m}^*_{i-1} \odot \boldsymbol{\theta}_0$ |
| w/ LR | $p$ | $\boldsymbol{s}^*_{i-1}$ | $\gamma$ | $\boldsymbol{m}^*_{i-1} \odot \boldsymbol{\theta}_0$ |
| (Default) Soft iEP w/ LR | $1-(1-p)^i$ | $\boldsymbol{s}^*_{i-1}$ | $\gamma$ | $\boldsymbol{\theta}_0$ |
| w/LR and w/ zero init | $1-(1-p)^i$ | $\boldsymbol{m}^*_{i-1} \odot \boldsymbol{s}^*_{i-1}$ | $\gamma$ | $\boldsymbol{\theta}_0$ |
| w/SRw and w/ zero init | $1-(1-p)^i$ | $\boldsymbol{m}^*_{i-1} \odot \boldsymbol{s}_0$ | $\gamma$ | $\boldsymbol{\theta}_0$ |

it is typical to rewind to the initial weight values saved at the first iteration, and this is a commonly used iterative method in IMP (Frankle et al., 2020).

**Score reinitializing (SRi)** Score reinitializing retrains by rewinding the learning rate, but the scores are reinitialized. Therefore, the initial values of the scores for the $i$-th cycle are randomly initialized $\tilde{\boldsymbol{s}} \sim \mathcal{D}_s$. The learning rate is the same as in learning rate rewinding and SRw.

**Learning rate rewinding (LR)** Learning rate rewinding retrains by rewinding only the learning rate. Although the initial values of the scores $\boldsymbol{s}$ in the $i$-th cycle are set to $\boldsymbol{s}_{i-1}$, the learning rate is reset to the initial value and decayed by the same scheduler as the first iteration.

Table 3 summarize the above-mentioned four variants of iEP. While these methods differ in what they inherit, they share a common aspect of keeping the masks from the previous cycle fixed. We denote this approach as *hard pruning* since it completely omits the bottom $p$ edges after each cycle.

### 4.2 DRAWBACK OF ITERATIVE HARD PRUNING AND ITS REMEDY

One immediate advantage of iterative hard pruning is that it allows for setting a lower pruning rate in each cycle. By completely removing the bottom $p$ percent of edges that were not selected in each cycle, achieving a desired pruning rate $k$ through multiple iterations of smaller pruning is possible. As shown in Table 1, the dying ratio decreases when the pruning rate is low, which seems reasonable at first glance. However, the above explanation does not consider the possibility that edges masked in the early stages were actually necessary, i.e., the existence of disguised dying edges. When performing hard pruning in this manner, there is a possibility that, in the worst-case scenario, we are generating false negative dying edges corresponding to the masks that have been removed so far.

We propose a new mechanism called *soft pruning* to alleviate the issue. Unlike hard pruning, soft pruning does not entirely disable the bottom $p$ edges at each cycle (Figure 9). Therefore, all edges can be selected at the end, regardless of whether it was chosen at former cycles. Similar to hard pruning, soft pruning also benefits from the lower pruning ratio, especially at the beginning of the iterative process. Specifically, at each cycle, $i$, soft iEP optimize Equation 1 inheriting scores of previous cycle $\boldsymbol{s}^*_{i-1}$ as the initial values of the scores. Note that, only LR rewinding and FT are applicable for soft pruning since if we rewind the score it degenerates to the standard EP. Since FT often provides worse performance, we use LR rewinding without mentioning otherwise.

It is worth noting that the proposed soft pruning is similar to the soft filter (channel) pruning (He et al., 2018; Kang and Han, 2020) proposed in pruning in dense networks. Specifically, unlike hard filter pruning which fixes the pruned filter, soft filter pruning assigns zero to the pruned weight at the pruning phase but the weight of the pruned filter is to be updated during subsequent optimization. Given that, we also compared different variants of soft pruning, which we denoted as soft pruning w/ zero init (the last row of Table 3). However, we found that initializing the score to zero at the pruning phase (the end of the cycle) induces the dying edge, and harts the performance (subsubsection C.2.1).

Since the primal difference between EP and soft iEP is whether the score is randomly initialized or pre-trained on prior cycles (see Table 3), the success of the soft iEP also indicates the importance of the initialization to reduce the dying edges. If the edges with equal importance (or uncertainty) have

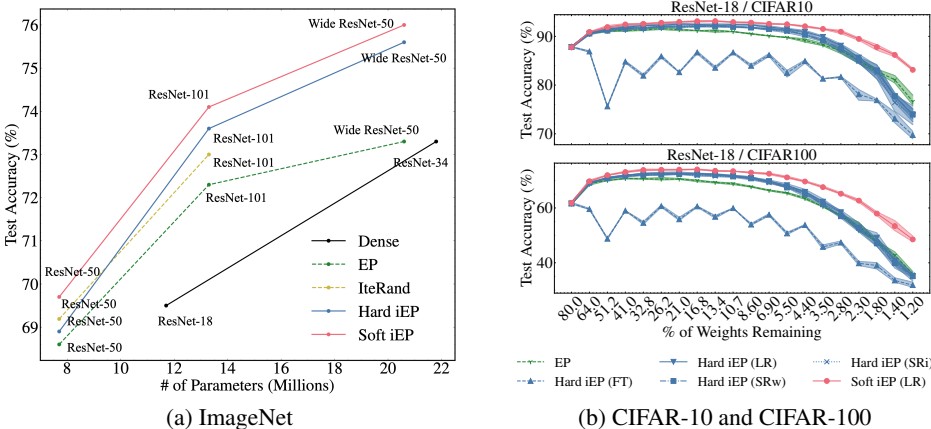

(a) ImageNet              (b) CIFAR-10 and CIFAR-100

Figure 3: Comparing the performance of EP, iEP w/ hard pruning, and iEP w/ soft pruning. (a) ImageNet, (b) CIFAR-10 and CIFAR-100. Both hard iEP and soft iEP use LR rewinding by default.

similar scores, the dying edge should be small since exchanging these edges becomes easier. The randomly initialized scores do not exhibit such cluster behavior. Most scores of the distribution of the scores leaned by soft iEP tended to be minimal values, while some took large values, i.e., soft iEP first identifies good edges and assigns similar scores on other edges that still need to be explored.

## 5 EXPERIMENTS

**ImageNet** Figure 3 (a) compared the subnetworks found by hard iEP (blue), soft iEP (red), EP (green), IteRand (Chijiwa et al. (2021), yellow), and dense networks trained by the standard gradient descent over weights (black). Regarding IteRand, we directly take the value from the original paper. They do not provide results on Wide ResNet50. Following (Ramanujan et al., 2020), the subnetworks are optimized using the SGD optimizer for 100 epochs with a learning rate of 0.3, weight decay of 0.00003, and momentum of 0.875. The learning rate is decayed using the cosine annealing after a 5-epoch for warm-up. The batch size is set to 512, and the weight initialization method is Signed Kaiming Constant. The number of iterations in iterative pruning is set to 3, and the percentage of weights remaining in the last iteration is set to 30% by decreasing $0.3^{\frac{1}{3}}$ at each iteration. We tested models with various sizes (ResNet-50/101 and Wide ResNet-50) for the edge-pop and iEP, and ResNet-18/34 for the standard weight training. For results other than iterative pruning, we cite the results of (Ramanujan et al., 2020; Chijiwa et al., 2021).

We can make the following observations. (1) The subnetworks discovered by soft iEP provide superior performance to all EP, IteRand, and trained dense networks across all model sizes. For example, with WideResNet-50 as the search spaces, soft iEP achieves 76.0% accuracy with about 20M parameters, which is about 2.7 points higher than EP and the dense training given a similar number of parameters. (2) The performance gain over EP expands as the model size increases. The results suggest that the standard EP can not handle large search spaces, but iEP (at least partially) cures the problem. (3) In iEP, it can be observed that soft pruning generally achieves higher performance than hard pruning.

Note that, while the Figure 3 (a) compares the Soft iEP and dense training given the same parameter count (thus same inference FLOPS), there are several differences in other aspects among them. For example, the dense training and EP require less training FLOPS since the proposed method employs iterative training. Besides, dense training is not tailored to sparse training and thus can be significantly improved by using advanced methods like LT (iterative magnitude pruning), RigL (Evci et al., 2020), dynamic sparse trainig (Liu et al., 2020), ITOP (Liu et al., 2021), Free Tickets (Liu et al., 2022), Sup Tickets (Yin et al., 2022), CigL (Lei et al., 2023), etc. To clarify this point, we compared the dense LT with the proposed method and its variants, showing that the dense training still has performance merits especially when we need higher sparsity (see subsubsection C.2.1). Nevertheless, the SLT-based method still has practical merits on the memory footprint as discussed in the introduction, encouraging future research to fill the gap between dense training and SLT. For example, 23M parameters (ResNet-50) can be stored as approximately 2.87M bytes (binary mask for

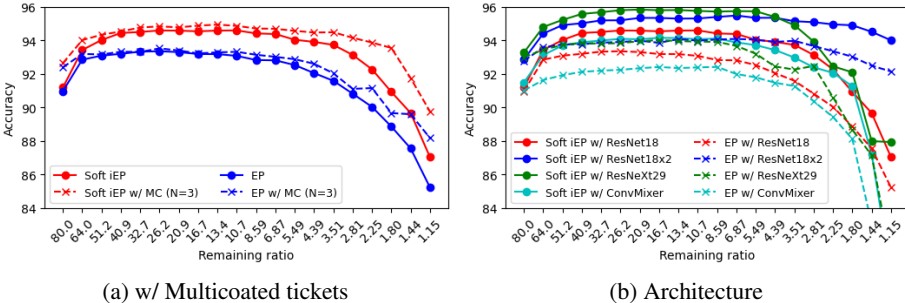

(a) w/ Multicoated tickets        (b) Architecture

Figure 4: (a) Comparing performance of soft iEP and multicoated lottery tickets (Okoshi et al., 2022). We use a 3-coated supermask both for EP and Soft iEP and use the linear option proposed in the original paper. (b) Soft iEP vs. EP with different architectures (ResNet18x2, ResNext, and ConvMixer). Results on CIFAR100 are provided in subsubsection C.2.2.

each edge), while dense models require 46M bytes (if we use the standard 32 bits for storing each edge). Structured pruning can further reduce the footprint of both SLT and dense networks.

**CIFAR-10 and CIFAR-100** Figure 3 (b) compares the performance of various retraining methods discussed in subsection 4.1. Here we show the results with ResNet18; the tendency is the same with other architectures (e.g., VGG). We used the SGD for 100 epochs with the cosine annealing LR scheduler. The batch size is set to 128, the learning rate to 0.1, the weight decay to 0.0001, and the momentum to 0.9. We set the shrinkage rate $p = 0.2$ at each cycle. Similar to Figure 3 (a), the colors correspond to different methods. We compare different rewinding techniques, namely fine-tuning (FT), learning rate rewinding (LR), score rewinding (SRw), and score re-initializing (SRi). We repeat the experiments with three seeds and report average performance and standard error. In addition, we compare the proposed method with multicoated tickets (Okoshi et al., 2022) in Figure 4-a, and compare EP and soft iEP on various backbone networks: ResNet18x2, ResNext(32x4d) (Xie et al., 2016), and ConvMixer(256x8) (Trockman and Kolter, 2022)).

We can make the following observations. (1) soft iEP stably performs best across datasets and remaining ratio, especially under the region of non-trivial remaining ratio (e.g., $< 0.035$). Hard iEP with learning rate rewinding performs slightly better than other rewinding methods, but its differences are marginal. The results suggest the merit of soft pruning. (2) Most retraining methods, except fine-tuning, provide performance gain over EP. When comparing the maximum test accuracy of ResNet-18, iterative pruning is 1.33 and 1.84 points higher than one-shot pruning on CIFAR-10 and CIFAR-100. (3) No rewinding (fine-tuning) performs poorly, even significantly worse than EP. The results are consistent with the prior knowledge of LT. (4) Multicoated tickets improve the performance on the high sparsity region on EP, but soft iEP generally outperforms it. Besides, Soft iEP can be combined with multicoated tickets, and provide similar performance gain. (5) Soft iEP stably improve the performance regardless of the architecture choice.

**Does soft iEP reduce the dying edges?** Figure 5-(a) compares the dying ratio when applying EP (green), hard iEP (blue), and soft iEP (red) on CIFAR-10. For hard iEP, we plot two values: (1) apparent dying edges (dashed line) that count the dying edges among $\boldsymbol{m}_{i-1} \odot \boldsymbol{\theta}_0$, and (2) worst case dying edges (solid line) that assume all masked edges are disguised dying edges. Note that the dying edges for iterative pruning are counted for each cycle, so it is fair to compare with EP and iEP. The figure shows that soft iEP drastically reduces the dying ratio in almost all regions. The dying ratio is near zero, especially at the beginning of a few cycles. Note that, while the dying ratio of soft iEP then increases and crosses with that of EP around 10%, the performance of soft iEP is significantly better than EP in the region, as shown in Figure 3-(b). Although it is hard to prove, the results suggest that soft iEP reduces the false negative dying edges, especially at the exploration phases (early cycles), and starts to remove true negative dying edges after a few cycles.

**Benefit of soft pruning** To deeply understand the benefit of soft pruning, we visualize the transition of score ranking during the entire cycle of iterative pruning ( Figure 5-(b)). Each line corresponds to the ranking of each edge (randomly sampled 200 edges), and red represents the edge selected at the 3.5% remaining ratio. By definition, all red edges are always above the threshold (black line) in hard pruning. On the other hand, the results of soft pruning show that the finally selected edge might

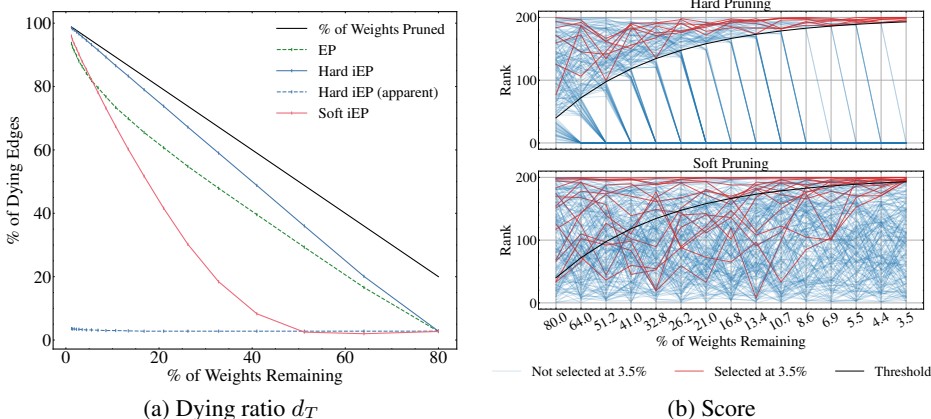

Figure 5: (a) Comparing the dying ratios at different remaining rates among various methods. (b) Visualizing the transition of score rank of randomly sampled 200 edges. (Red) selected at 3.5% remaining rate, (blue) not selected at the remaining ratio, and (black) threshold of each cycle.

have smaller scores in earlier cycles. In other words, the results suggest that the scores might not be well calibrated and should not be entirely discarded, especially in the earlier cycle of iterative pruning and should not be entirely discarded, especially in the earlier cycle of iterative pruning. In subsection C.3, we further discuss the difference of subnetworks obtained by EP, Hard iEP, and soft iEP, and show that they converge to different structures and soft iEP use parameters more effectively.

## 6   CONCLUSION AND LIMITATION

**Empirical analysis of EP from the dying ratio perspective.**   In section 3, we introduce the notion of the dying ratio that measures the exploration inefficacy to find a good structure. Table 1 show that EP induces surprisingly many dying edges, and Table 1 suggest that the higher dying ratio hinder the performance. Results provide new insight into the behavior of EP, which needs to be better investigated and led to the invention of a better algorithm.

**Testing various iterative pruning in SLT for the first time.**   While the benefit of LT is well-known, its benefit is not empirically investigated in SLT. We tested variants of iterative pruning customized for SLT (summarized in Table 3), and tested it on standard benchmarks (Figure 3). The results might be a starting point for discussing the connection between LT and SLT from the methodological perspective and investigating why iterative pruning is necessary to find (strong) lottery tickets.

**Contribution 3: Proposal of soft pruning technique to improve performance.**   In addition to the direct application of iterative pruning, we propose a new mechanism tailored for SLT setup (Figure 9. Figure 5-(a) shows that our proposal successfully reduces the dying edges, and Figure 3 shows that our proposal stably outperforms all baselines. Notably, the subnetwork discovered by soft iEP from WideResNet-50 achieves 76.0% accuracy, which is 2.4 points higher than EP, and the dense training gave a similar number of parameters and outperformed hard iEP and IteRand.

**Limitation**   (1) The proposed method is the increased search time since we apply EP iteratively. This issue is not unique to SLT and is discussed in the conventional LT. Existing research tries to reduce retraining steps after rewinding, which could also be applied to our setup. (2) The theoretical investigation of why dying edges occur and the theoretical justification for the success of soft pruning would be the next step worth investigating and validating. Besides, the connection between the dying edge problem and prior theoretical discussions is worth investigating. For example, Pensia et al. (2020) proved the existence of SLT using a subset-sum formulation, i.e., they show that a subset-sum of uniformly selected values can approximate the weight of the target network. In one aspect, our findings support the theoretical analysis, as the proposed method improves performance by enlarging the search space. However, it is also worth exploring why the original EP still works sufficiently well even with a very high dying ratio, which contradicts the assumption of the subset sum formulation.

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
