# A    DETAIL SETUP OF SECTION 3

## A.1    TRANSITION OF $d_t$ DURING TRAINING

Figure 6 illustrates the transition of dying ratio $d_t$ during edge-popup training on CIFAR-10. Specifically, we applied edge-popup (Equation 1) with various pruning ratios $k$ (20% to 83.2%) to ResNet-18, whose weights are initialized by the Signed Kaiming Constant (Ramanujan et al., 2020). We followed the configurations of (Ramanujan et al., 2020). Namely, we used SGD optimizer for 100 epochs with batch size 128, weight decay 0.0001, momentum 0.9, and learning rate 0.1, which is decayed using a cosine annealing learning rate schedule. Scores are initialized using Kaiming Uniform initialization.

As shown in this figure, regardless of $k$, the change in the early stage of training is large, and the ratio decreases by about 15-20% and then becomes constant. For example, the dying ratio of the green line ($k = 0.488$) starts around $d_0 = 0.488$, then decreases to around $d_T = 0.38$. Since approximately half (48.88%) of edges are finally masked in these configurations, the results indicate that about 77% of masked edges are actually never selected. While its starting dying ratio naturally depends on the prune ratio $k$, the tendency is almost the same. Since the subnetworks are first selected entirely random at the first iteration, the results suggest that EP might be trapped around the local minima nearby random subnetworks and can not effectively search the entire spaces of subnetworks.

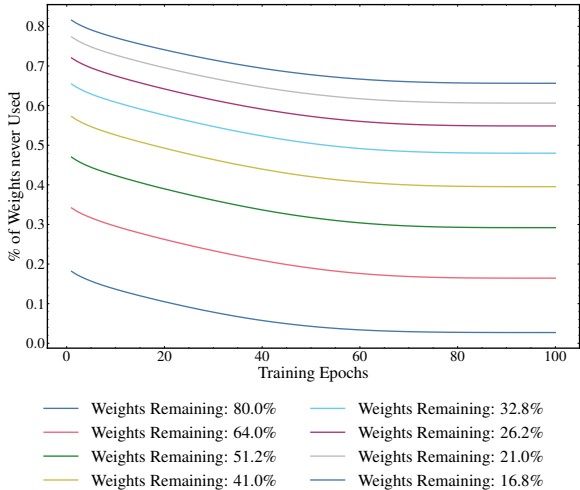

Figure 6: Transition of the dying ratio on various pruning ration.

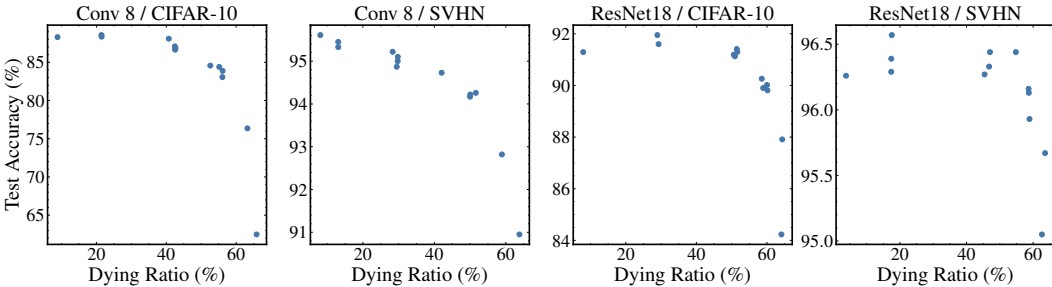

Figure 7: The relationship between test accuracy and dying ratio when applying EP to Conv 8/ResNet18 on CIFAR-10/SVHN with various hyperparameters.

## A.2 DYING RATIO WITH VARIOUS HYPERPARAMETERS

Table 4: The effect of various hyperparameters on test accuracy (Test Acc) and dying ratio (DR) is shown. As default parameters, we use (batch size, learning rate, weight decay, optimizer, score initialization, weight initialization, warmup steps) = (128, 0.1, 1e-4, SGD, Kaiming Uniform (uniform), Signed Kaiming Constant (sc), 0), and the pruning rate is set at 0.7 in all experiments. Each row represents the results when only a specific parameter is changed to the value of a candidate.

| | | Conv8 | | | | ResNet18 | | | |
| | | CIFAR-10 | | SVHN | | CIFAR-10 | | SVHN | |
| Variable | Candidate | Test Acc | DR | Test Acc | DR | Test Acc | DR | Test Acc | DR |
|---|---|---|---|---|---|---|---|---|---|
| (Base line) | | 86.65 | 42.52 | 95.10 | 29.78 | 91.13 | 50.93 | 96.29 | 17.39 |
| Batch size | 64 | 88.35 | 21.49 | 95.33 | 13.11 | **91.95** | 28.94 | 96.39 | 17.39 |
| | 128 | - | - | - | - | - | - | - | - |
| | 256 | 83.06 | 56.06 | 94.17 | 49.99 | 89.81 | 60.15 | 96.13 | 58.71 |
| | 512 | 76.36 | 63.22 | 92.82 | 58.91 | 87.91 | 64.33 | 95.67 | 63.59 |
| Learning rate | 0.05 | 83.88 | 56.12 | 94.22 | 50.05 | 90.02 | 60.03 | 96.16 | 58.64 |
| | 0.1 | - | - | - | - | - | - | - | - |
| | 0.2 | **88.56** | 21.43 | 95.45 | 13.10 | 91.60 | 29.29 | **96.57** | 17.56 |
| | 0.4 | 88.29 | **8.86** | **95.61** | **8.10** | 91.29 | 7.92 | 96.26 | 3.81 |
| Weight decay | 0.0 | 84.4 | 55.16 | 94.26 | 51.60 | 89.90 | 58.92 | 95.93 | 58.90 |
| | 1e-4 | - | - | - | - | - | - | - | - |
| | 1e-2 | 39.52 | 24.87 | 19.90 | 29.29 | 85.93 | **0.01** | 95.68 | **0.00** |
| Optimizer | SGD | - | - | - | - | - | - | - | - |
| | LARS | 62.48 | 65.84 | 90.95 | 63.80 | 84.23 | 64.10 | 95.05 | 62.64 |
| Score init | k-uniform | - | - | - | - | - | - | - | - |
| | k-normal | 84.57 | 52.60 | 94.73 | 42.04 | 90.26 | 58.53 | 96.44 | 54.82 |
| Weight init | sc | - | - | - | - | - | - | - | - |
| | scaled sc | 88.09 | 40.70 | 95.22 | 28.33 | 91.19 | 50.61 | 96.27 | 45.41 |
| Warmup steps | 0 | - | - | - | - | - | - | - | - |
| | 5 | 86.84 | 42.64 | 95.00 | 29.74 | 91.41 | 51.44 | 96.33 | 46.78 |
| | 10 | 87.09 | 42.46 | 94.87 | 29.47 | 91.30 | 51.61 | 96.44 | 47.04 |

**Dying ratio negatively correlates with the performance.** While Table 1 already indicates the low exploration efficacy of EP, it is still unclear whether it affects the final performance. To roughly investigate the relationship between the dying ratio and the final performance, we optimize EP with different hyperparameters, including batch size (64, 128, 256, and 512), learning rate (0.05, 0.1, 0.2, 0.4), weight decay (0.0, 1e-4, and 1e-2), optimizer (SGD and LARS), score initialization (Kaiming-uniform and Kaiming-normal), weight initialization (signed constant and scaled signed constant), and warm up steps (0, 5, and 10). We fix the pruning rate $k = 0.7$. The detailed configurations and full results are shown in Table 4 ( Appendix A). Figure 7 plots the relationship between final

test accuracy and the dying ratio. The results show that for Conv architecture on both CIFAR-10 and SVHN, the dying ratio and the final performance exhibit a negative correlation. In ResNet18, it exhibits an inverse U-shape, suggesting that a high dying ratio harts performance. Specifically, some hyperparameters, including large learning rates or equivalently small batch sizes, drastically reduce the dying ratio on both Conv8 and ResNet18. On the other hand, some parameters do not decrease the dying ratio as much, such as score initialization, weight initialization, and the type of optimizers. In addition, as the dying ratio decreases, the final test accuracy is also slightly increased. However, changing such parameters also might induce optimization difficulty and not be desirable in some cases. For example, with higher weight decay, the edge-pop often converges to useless subnetworks. It is worth noting that there might be a hidden covariate between the dying ratio and the performance, e.g., the stability of the optimization. Therefore, the results do not indicate the causality between the dying ratio and the performance but suggest that the high dying indicates a suboptimal solution.

**Why hyper parameter matter for the dying ratio?**     As described in the section 2, EP optimizes scores using the gradients. Formally, if $\theta_{uv} z_u$ denote the weighted output of neuron $u$, and $x_v$ denote the input of neuron $v$ ($z_v = \sigma(x_v)$ where $\sigma$ is an activation function), EP updates $s_{uv}$ as

$$s_{uv}^t \leftarrow s_{uv}^{t-1} - \gamma \frac{\delta L}{\delta x_v} \theta_{uv} z_u, \tag{3}$$

where $\gamma$ is a learning rate. Assume $\Phi^t$ denote the index of top-k edges given the score vector $\mathbf{s}$. Edge swapping only occurs if there exists $i \notin \Phi^{t-1}, j \in \Phi^{t-1}$ that meet $(s_i^t > s_j^t) \wedge (s_i^t < s_j^t)$.

Obviously, the selection of the hyperparameter affects the occurrence of the edge swapping and thus the dying edge. For example, if we use a small learning rate, edge swapping will be less likely to occur. Equivalently, if we use a large batch size (Smith et al., 2018), then it also prevents weight swapping as shown in Table 4. However, the choice of the learning rate and batch size often causes optimization difficulty, which also affects the performance, and thus another mechanism to increase the chance of edge swapping is necessary to prevent the dying edges.

Instead of making the step size large by using large learning rate or smaller batch size, one can decrease the magnitude of the scores to enhance edge swapping. This is a reason why higher weight decay decreases the dying ratio Table 4. However, using higher weight decay makes all scores smaller regardless of whether it is actually important, and thus might cause performance drops.

### A.3 SHARED WEIGHTS AMONG DIFFERENT SEEDS

Figure 8 visualize the ratio of mask matrix $m$ that is shared across different seeds, starting from the same initial weights and scores but in different order of batches. Specifically, given two masks $m_1$ and $m_2$, we count the elements where $m_1 - m_2$ is zero. Since it starts from identical scores and weights, the initially shared ratio is always $100\%$. We used prune rate $k = 0.2$, and applied edge-pop on ResNet18 using CIFAR-10. Other configurations are the same with Equation 3.

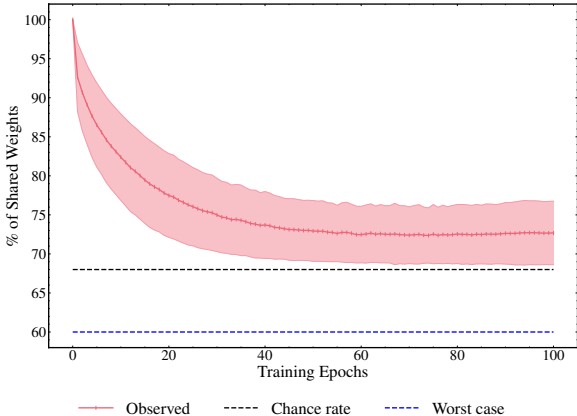

Figure 8: Visualizing the ratio of mask matrix $m$ that is shared across different seeds, starting from the same initial weights and scores but in different order of batches. (Red) average shared ratio across layers, (black) the chance ratio ($= k^2 + (1 - k)^2 = 0.68$), and (blue) the worst case ($=1 - 2 \times k = 0.6$).

## B  DETAIL ABOUT PROPOSAL

### B.1  PSEUDO CODE OF SOFT IEP WITH LR REWINDING.

---

**Algorithm 1** Entire procedure of iterative soft pruning.

---

**Require:** Random networks $\boldsymbol{\theta}_0$ (from signed constant distribution by default)
**Require:** Random scores $\boldsymbol{s}_0$ (from uniform distribution by default)
**Require:** shrinkage rate $p$, the number of cycle $M$, and learning rate $\gamma$
1: $\boldsymbol{s} \leftarrow \boldsymbol{s_0}$
2: **for** $i \leftarrow 1$ to $M$ **do**
3:     $k_i \leftarrow 1 - (1 - p)^i$
4:     **while** convergence **do**
5:         update $\boldsymbol{s}$ using equation 1 with $k = k_i$.
6:     **end while**
7:     $\boldsymbol{s}_i^* \leftarrow \boldsymbol{s}$
8:     $\boldsymbol{m}_i^* \leftarrow \boldsymbol{m}_{k_i}^{\boldsymbol{s}}$
9:     Reset learning rate # LR rewinding
10:     **if** use zero init option **then**
11:         $\boldsymbol{s} \leftarrow \boldsymbol{s}_i^* \odot \boldsymbol{m}_i^*$
12:     **end if**
13: **end for**

---

Algorithm 1 summarize the entire algorithm of the Soft iEP. When using the Hard iEP, we freeze the pruned weights at the end of each cycle. See Table 3 for the comparison of the methods.

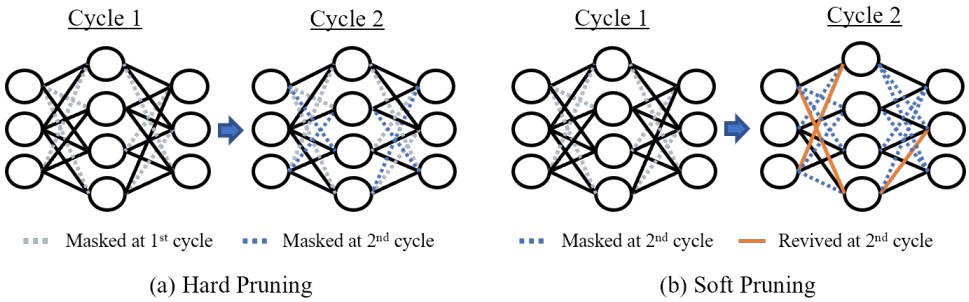

Figure 9: Comparison of hard pruning and soft pruning (proposal).

## B.2 HARD PRUNING VS. SOFT PRUNING

Figure 9 illustrate the difference between soft pruning and hard pruning. See section 4 for the detailed description.

## C ADDITIONAL EXPERIMENTS

### C.1 EP WITH PROGRESSIVELY INCREASED PRUNING RATE

Table 5: EP with progressively increased pruning rate. The total training time is fixed at 100 epochs, and the pruning rate is gradually increased during training #cycle (1, 2, 4, 8, 16) times so that the final remaining rate is 0.328 or 0.012. EP is corresponding to the #cycle=1. Lin refers to the linear pruning rate scheduling, and exp refers to the exponential scheduling. For reference, we also included the results of soft iEP.

|          | 0.328 |       | 0.012 |      |
|----------|-------|-------|-------|------|
| #cycle   | lin   | exp   | lin   | exp  |
| 1        | 91.3  | 91.3  | 76.6  | 76.6 |
| 2        | 89.2  | 89.1  | 74.0  | 74.0 |
| 4        | 90.4  | 90.79 | 68.4  | 75.4 |
| 8        | 90.6  | 90.82 | 55.1  | 74.2 |
| 16       | 90.5  | 90.83 | 37.0  | 76.6 |
| Soft iEP | 92.6  | 92.6  | 83.1  | 83.1 |

We tested EP with a progressively increased pruning rate. Specifically, while fixing the total computation costs (100 epochs), we gradually increase the pruning rate #cycle times during training. Results show that the strategy does not perform better than EP (#cycle=1.0), suggesting that progressively increasing the pruning rate is insufficient and iterative pruning is necessary.

### C.2 ADDITIONAL COMPARISON

In this section, we provide the following additional experiments: (1) a comparison with dense lottery tickets to discuss the performance gap between SLT and dense training, (2) a comparison among different soft pruning strategies, (3) a comparison with an existing SLT method (multicoated tickets) to further clarify the benefit of soft iEP, (4) applicability of soft iEP to diverse architecture, and (5) applicability of soft iEP to the pruning of pre-trained weights (following (Zhang et al., 2023)). We use CIFAR-10 and CIFAR-100 as datasets, and ResNet18 as the default architecture. Experimental configurations are the same as in the section 5.

#### C.2.1 DENSE LT AND SOFT iEP WITH ZERO INIT

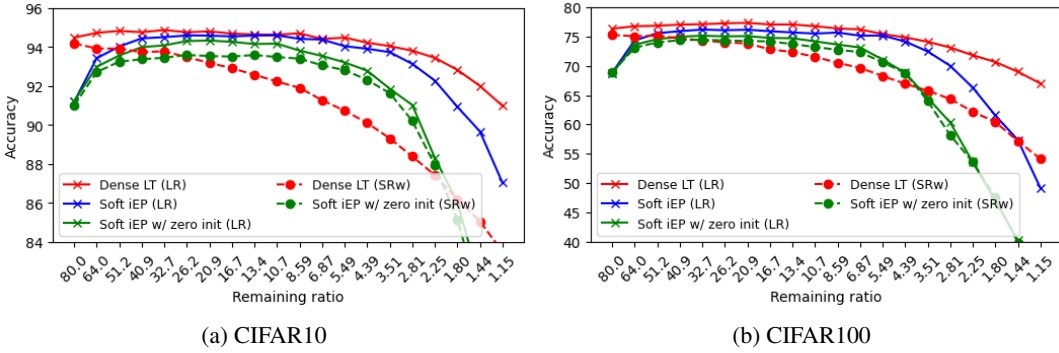

(a) CIFAR10                     (b) CIFAR100

Figure 10: Comparing performance of EP and Soft iEP with different architectures.

Figure 10 compares the Dense LT (red line), Soft iEP (blue line), and Soft iEP w/ zero initialization, which makes the score corresponding to the pruned weight zero at the end of the cycle as shown

in Table 3. We can make the following observations. Firstly, the results show that dense LT w/ LR rewinding still performs better than soft iEP, especially on the high sparsity region, given the same parameter count (inference FLOPS). Secondly, soft pruning w/ zero initialization does not work well, providing significant performance degradation. This might be because, once we make the score zero, it is less chance to be re-activated due to the exploration inefficiency of the gradient-based method as we discussed in the paper.

### C.2.2  COMPARISON WITH MULTICOATED LOTTERY TICKETS

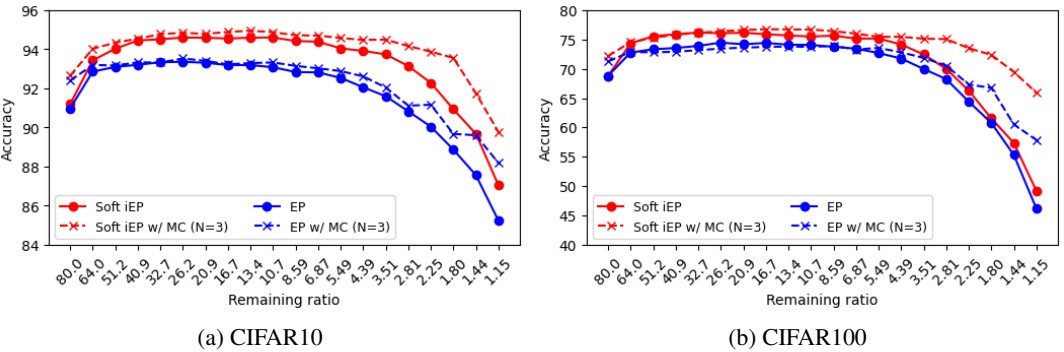

(a) CIFAR10                      (b) CIFAR100

Figure 11: Comparing performance of soft iEP and multicoated lottery tickets (Okoshi et al., 2022). We use a 3-coated supermask both for EP and Soft iEP and use the linear option proposed in the original paper.

Figure 11 compare the multicoated lottery tickets proposed by (Okoshi et al., 2022) and proposed methods. Multicoated lottery tickets use multiple masks corresponding to the different remaining ratios to make a prediction. To be more specific, given a mask vector $m_s^{k_1}, m_s^{k_2}, \cdots m_s^{k_N}$, where $k_1 > k_2 > \cdots > k_N$, multicoated lottery tickets multiply weights with sum of the mask:

$$W = \sum_{i \in 1 \cdots N} m_s^{k_i} \odot \theta, \tag{4}$$

and $W$ to make a prediction. We use $N = 3$ and linear spanning introduced in (Okoshi et al., 2022) to choose the $k_i$.

The results show that multicoated tickets generally improve performance, especially in high-sparsity regions. The extension can be mixed with our proposal, which stably provides the best performance.

### C.2.3  ADDITIONAL ARCHITECTURES

Figure 12 compare the EP (dashed line) and Soft iEP (solid line) with four different architectures: ResNet18 (default), ResNet18x2, ResNext(32x4d) (Xie et al., 2016), and ConvMixer(256x8) (Trockman and Kolter, 2022)). Different colors correspond to the different architecture. On both datasets, Soft iEP stably increases the performance regardless of the choice of architecture. With ConvMixer and CIFAR100, both EP and Soft iEP do not perform well. Among network architecture, ResNext(32x4d) generally performs better on the low-sparsity region, but ResNet18x2 performs well on high high-sparsity region (with both EP and soft iEP).

### C.2.4  DIFFERENT WEIGHT INITIALIZATION

Figure 13 compare the performance on different weight initialization. Specifically, Figure 13 compares the performance when we use kaiming normal distribution to initialize weight and apply Hard iEP and Soft iEP (dashed blue and red). The results show that when the weights are initialized from a normal distribution and thus do not a constant values, soft iEP and hard iEP provide on-par performance.

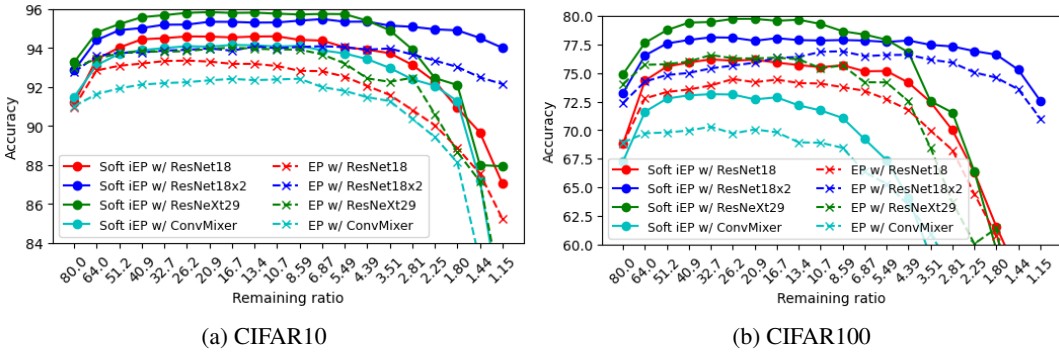

(a) CIFAR10

(b) CIFAR100

Figure 12: Comparing performance of EP and Soft iEP with different architectures.

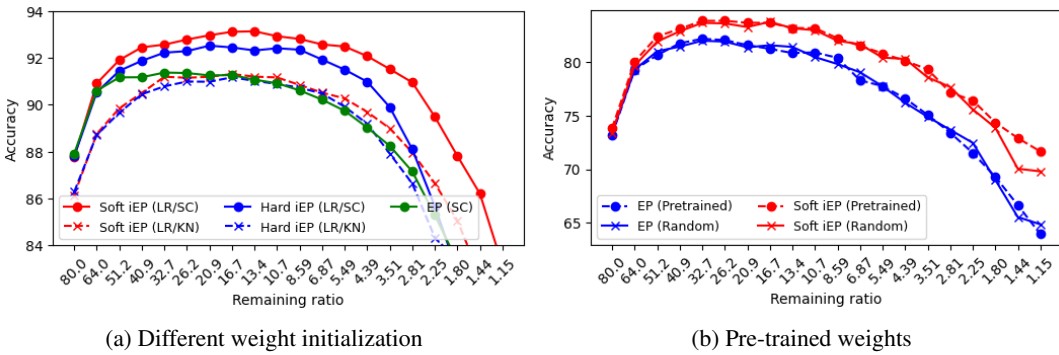

(a) Different weight initialization

(b) Pre-trained weights

Figure 13: Comparing performance of EP and Soft iEP with different weights initialization. (a) Comparing different randomization (SC: signed constant, KN: kaiming norm). (b) Comparing pre-trained weights and random weight (signed constant).

Following (Zhang et al., 2023), we also tested the applicability of Soft iEP to prune pre-trained weights ( Figure 13-b). Specifically, we use pre-trained ResNet18 provided by timm library (Wightman, 2019). The results show that Soft iEP generally performs better than EP, but at the same time, Soft iEP w/ pre-trained weights provide on-par performance with Soft iEP w/ random weights, meaning that the iEP still does not use any knowledge embedded in the pre-trained model.

### C.2.5 CONVERGENCE SPEED

Figure 14 compare the speed of convergence between Hard iEP and Soft iEP. As a result, we can see that Hard iEP provides better performance at the beginning of the training, may be due to it optimizing fewer parameters, but Soft iEP finally outperformed it.

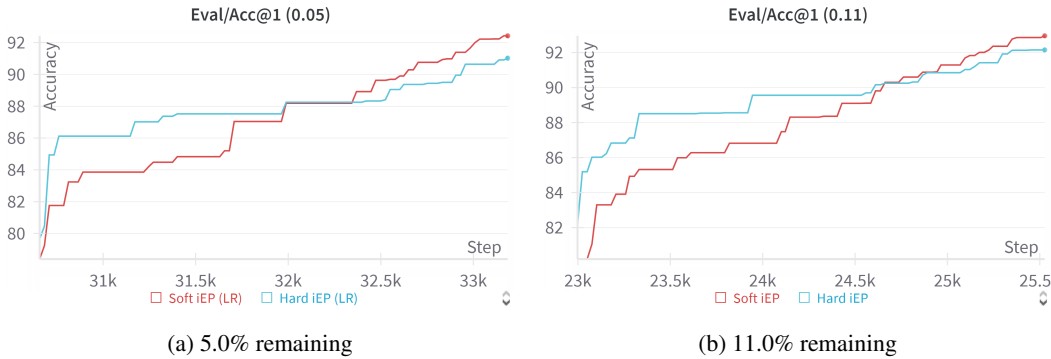

(a) 5.0% remaining        (b) 11.0% remaining

Figure 14: Comparing convergence speed of Hard iEP and Soft iEP when using different remaining ratios.

## C.3 ANALYZING THE STRUCTURE OF THE SUB-NETWORKS OBTAINED BY DIFFERENT ALGORITHMS

### C.3.1 JACCARD DISTANCE

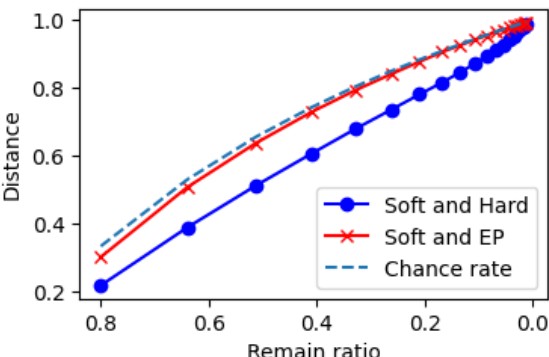

Figure 15: Comparing the structure obtained by soft iEP, hard iEP, and EP. The dataset and architecture are CIFAR10 and ResNet18 respectively. The blue line represents the Jaccard distance between Soft iEP and Hard iEP at each remaining ratio, and the red line represents the distance between Soft iEP and EP. The dashed line represents the chance ratio, which can be calculated by $k^2/(2k - k^2)$.

### C.3.2 DYING NEURONS

Table 6: Analyzing the structure difference from the perspective of the dying neuron of the second last layer. Specifically, the first block (input side) represents the number of neurons in the second last layer (512 neurons) that have no connection from the prior neurons as a consequence of training. Similarly, the second block (output side) represents the number of neurons that have no connection to the output neurons. The third block represents the ratio of the neuron that has no outside connection and input side connection. See Figure 16 for the illustration of the analysis.

|  | $k$ | 0.107 | 0.035 |
|---|---|---|---|
| No input-side connection | EP | 0 | 213 |
|  | Hard iEP (LR) | 14 | 157 |
|  | Soft iEP (LR) | **92** | **326** |
| No output-side connection | EP | **191** | **393** |
|  | Hard iEP (LR) | 89 | 331 |
|  | Soft iEP (LR) | 94 | 331 |
| Overwrap ratio | EP | 0% | 54% |
|  | Hard iEP (LR) | 15% | 47% |
|  | Soft iEP (LR) | **98%** | **98%** |

Figure 15 compare the structure similarity of the masks trained by Soft iEP, Hard iEP, and EP. Following (Paganini and Forde, 2020), we use the Jaccard distance:

$$d_j(\boldsymbol{m_1}, \boldsymbol{m_2}) = 1 - \frac{|\boldsymbol{m_1} \wedge \boldsymbol{m_2}|}{|\boldsymbol{m_1} \vee \boldsymbol{m_2}|}. \tag{5}$$

As shown in the figure, EP and Soft iEP learn completely different structures same as chance rate. Hard iEP and Soft iEP provide slightly similar structures, but it is getting different as we repeat the pruning. But how does the structure differ in more detail?

To analyze the in-depth difference among structures, we count the dying neuron that exists in the second last layer (see Figure 16). Specifically, we count the number of neurons that have no input-side

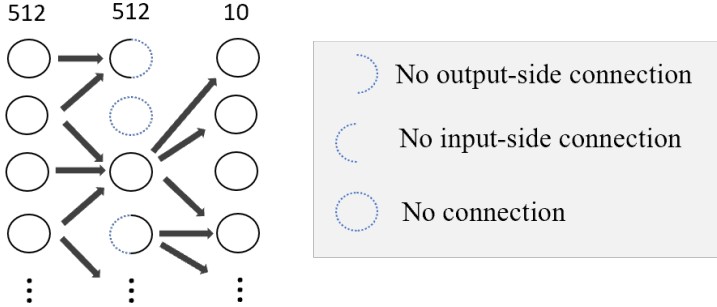

Figure 16: Illustration of the analysis of Table 6.

connection (first block of Table 6) and neurons that have no output-side connection (first block of Table 6). Results show that EP and Hard iEP provide a lot of neurons that have no output-side connection but still have an input-side connection. Since the neuron does not affect the prediction, the edges connected to such a neuron must be useless. Contrary, in the case of Soft iEP, most neuron with no output-side connection also does not contain any input-side connection (92/94 and 326/331), which means that the Soft iEP assign the scores to potentially more useful edges.

### C.3.3 COUNTING THE NUMBER OF VISITS

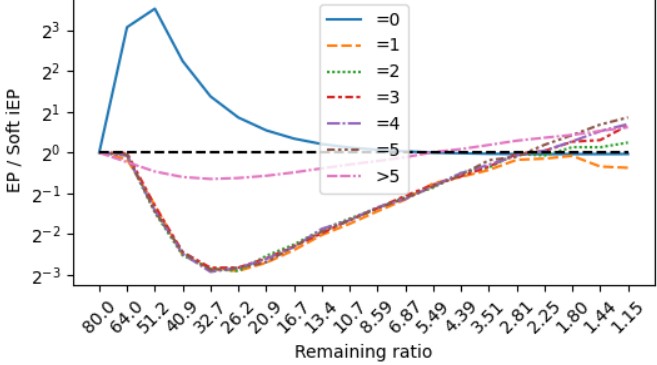

Figure 17: Counting the number of visits of each edge. The graph shows the ratio of EP / Soft iEP. The blue line shows the ratio of non-visit edges, which corresponds to the definition of the dying edges. The other line indicates the count of edges visited only once, twice, three times, four times, five times, and more than that. The black dashed line is an auxiliary line ($y = 2^0 = 1.0$).

While in the main paper, we have mainly discussed the dying edges, which have never been activated during training, one can also examine the visit counts of each edge, i.e., the count of the neurons activated during training. Figure 17 compare the count between EP and Soft iEP. For example, the blue line represents the EP contains more than $2^3$ times more dying edgest than Soft iEP. Instead, Soft iEP tends to activate neurons a few times to evaluate the neuron.

## D    MISC

### D.1    CODE

Code is available at `https://anonymous.4open.science/r/iterative_edgepop`

### D.2    TOTAL AMOUNT OF COMPUTE

We run our experiments mainly on cloud V100 or A100 instances. We used approximately 16 hours for training a ResNet-18 model for 100 epochs $\times$ 20 cycles on CIFAR-10/100 with the V100x1 instance, and 11/16/20.5 hours for training a ResNet-50/101/WideResNet-50 model for 100 epochs $\times$ 3 cycles on ImageNet with the A100x8 instances.

### D.3    LICENSE OF ASSETS

**Datasets**    We use the following 4 datasets: ImageNet (non-commercial research and educational purposes), CIFAR-10/100 (unspecified), and SVHN (non-commercial research and educational purposes).

**Codes**    Our codebase is mainly based on `https://github.com/allenai/hidden-networks` (Apache-2.0).