# OpenReview forum: "Soft iEP: On the Exploration Inefficacy of Gradient Based Strong Lottery Exploration"
_ICLR.cc/2024/Conference — Submitted to ICLR 2024_

### Official Review · Reviewer_tFae · 2023-10-28

**Soundness:** 2 fair
**Presentation:** 3 good
**Contribution:** 1 poor
**Rating:** 3
**Confidence:** 5

**Summary:**

Summary:
To tackle the inefficient dying edge phenomenon when training a sparse model with the strong lottery ticket hypothesis, this paper proposes a soft iterative edge-pop-up to explore possible edges that are masked earlier in training with an iteratively increasing pruning ratio. Experiments on training Resnet model families on ImageNet and CIFAR show that soft iEP improves EP and sometimes even outperforms the dense counterpart.

**Strengths:**

Pros:
1. Good preliminary study to intuitively show the widely existing dead edge problems and detrimental high dying ratio to pruning performance.
2. The writing is clear and easy to follow.

**Weaknesses:**

Cons:
1. Need comparison to some "drop and then grow" pruning methods in LTH like Rigging the Lottery and its following works. In the related studies, the authors explained that the difference between SLTH and LTH is that SLTH assumes there are strong subnets in dense models without training. However, the methods and experiments involve model training. Therefore, it is necessary to compare dynamic sparse training methods in LTH.
2. Soft pruning was directly added to iEP and the best-performing learning rate rewinding setting during the experiment. Need ablation studies: (1) decompose soft pruning and iEP to examine which contributes most; (2) combine soft pruning with different variants of iterative pruning to demonstrate its effectiveness.
3. Similar to cons 1, related studies compared in the experiment only include IteRand about iterative pruning. And IteRand is not implemented with the Wide ResNet-50 setting.
4. Figure 6(b) is difficult to comprehend, as the x-axis and the red and blue lines all refer to the weight remaining rate.
5. DST based approaches like RigL and ITOP should serve as necessary baselines for empirical comparisons.
6. The comparison in Figure 5 is unfair. Although the dense network (ResNet18) has similar parameters counts with a sparse ResNet50, the former needs much less training time since the proposed sparsity is unstructured and hard for acceleration. More discussions are needed to avoid misleading.
7. Also, the dense baseline performance is weak. Based on previous literature like https://arxiv.org/pdf/2210.04092.pdf, the dense ResNet-18 on CIFAR10 and CIFAR100 can reach ~95% and ~77% respectively, which are ~5% and ~17% accuracies better then the ones in Figure 5.

**Questions:**

Refer to the weakness section.

---

> ### Author Response · Authors · 2023-11-23
> **Response**
>
> Thank you for your insightful comments. We will answer your questions to address your concerns. Please also check the general response, in which the revision during the rebuttal period is explained.
>
> > 1. Need comparison to some "drop and then grow" pruning methods in LTH like Rigging the Lottery and its following works. In the related studies, the authors explained that the difference between SLTH and LTH is that SLTH assumes there are strong subnets in dense models without training. However, the methods and experiments involve model training. Therefore, it is necessary to compare dynamic sparse training methods in LTH.
>
> Thank you for providing interesting literature. Firstly, we admit that a performance gap exists between weight-based LT and SLT, and the paper should discuss this point more. Therefore, we have included a detailed discussion on the drawbacks of the SLT-based method compared to the weight-based LT in section 5. Besides we also included a comparison with weight-based LT in Appendix C.2.
>
> However, we would like to clarify again that SLTH still differs from LTH in the sense the weights are never updated in SLTH, which makes the SLTH unique in both intriguing properties of random neural networks and practical benefits. For practical benefit, please see the general response. Therefore, we believe that the performance gap between weight-based LT and SLT does not hinder the value of the paper. In addition, to strengthen the paper, we added an SLTH-based baseline (see general response 2-3).
>
> > 2. Soft pruning was directly added to iEP and the best-performing learning rate rewinding setting during the experiment. Need ablation studies: (1) decompose soft pruning and iEP to examine which contributes most; (2) combine soft pruning with different variants of iterative pruning to demonstrate its effectiveness.
>
> Thank you for the comments. However, since the difference between Hard iEP w/ LR and Soft iEP w/ LR is whether it employs hard or soft pruning, I’m afraid I think the effect of soft pruning and rewinding strategy is already decomposed in experiments. Besides, since soft pruning w/ SRw degenerated to the standard EP, we can think of the difference between EP and Soft iEP w/ LR as the difference in rewinding strategy.
>
> >3. Similar to cons 1, related studies compared in the experiment only include IteRand about iterative pruning. IteRand is not implemented with the Wide ResNet-50 setting.
>
> We directly obtained the value from the original manuscript, which does not provide results of Wide ResNet-50. We have clarified that in the revised manuscript. Due to the lack of enough computational budget (A100 specifically), we can not conduct the experiments on ImageNet during the rebuttal.
>
> > 4. Figure 6(b) is difficult to comprehend, as the x-axis and the red and blue lines all refer to the weight remaining rate.
>
> Sorry for the confusion. I have updated the explanation. To clarify, blue lines do not refer to the weight remaining ratio. It represents the transition of score ranking of each edge. Color represents whether the edges are finally selected or not. See the end of the P8 in the revised manuscript.
>
> > 5. DST based approaches like RigL and ITOP should serve as necessary baselines for empirical comparisons.
> > 6. The comparison in Figure 5 is unfair. Although the dense network (ResNet18) has similar parameters counts with a sparse ResNet50, the former needs much less training time since the proposed sparsity is unstructured and hard for acceleration. More discussions are needed to avoid misleading.
>
> Please check our response to the first comments.
>
> > 7. Also, the dense baseline performance is weak. Based on previous literature like https://arxiv.org/pdf/2210.04092.pdf, the dense ResNet-18 on CIFAR10 and CIFAR100 can reach ~95% and ~77%, respectively, which are ~5% and ~17% accuracies better than the ones in Figure 5.
>
> Thank you for carefully reading the paper and providing detailed comment, but I’m afraid the reviewer might have some sort of misunderstanding. The baseline with a lower prune ratio performs on par with the provided results, and Fig 5. of CIFAR10 does not provide a dense baseline.

---

### Official Review · Reviewer_He2e · 2023-10-31

**Soundness:** 3 good
**Presentation:** 3 good
**Contribution:** 2 fair
**Rating:** 5
**Confidence:** 4

**Summary:**

This paper empirically identifies that the edge-popup technique yields suboptimal performance because of the existence of a high dying edge ratio resulting from poor exploration of the  search space. To tackle this, the paper proposes a Soft iEP technique that iteratively prunes the subnetwork from the initialized network based on the Edge-Popup algorithm. The proposed technique keeps the chance for the bottom edges to be selected in each cycle and thereby effectively explores search space leading to a lower dying edge ratio. The experimentation conducted on multiple datasets showcases the effectiveness of the proposed technique.

**Strengths:**

* The motivation behind proposing the Soft iEP is well justified with the help of multiple empirical evidence. Also, the authors have done a good job in terms of empirically identifying the problem of dying edge in the Edge-Popup (EP) algorithm.
* The experimentation is conducted on a wide range of datasets ranging from easy datasets (e.g., Cifar10) to difficult datasets (e.g., ImageNet).
* The superior performance of the proposed Soft iEP is very convincing and intuitive.
* The paper is well written with the help of multiple visualizations. Also, the writing is very coherent and easy to follow.

**Weaknesses:**

* One of the reasons for having increased popularity of the EP algorithm is its computational efficiency compared to iterative techniques. Specifically, without iterative pruning, we can easily get the desired subnetwork (winning ticket) in one-shot training. The proposed technique misses the key advantage of the EP algorithm as the proposed Soft iEP requires iterative pruning. Therefore, the proposed technique may be computationally expensive and may limit its applicability in crucial domains such as large language models (LLM) where the computational cost is very expensive.
* The proposed techniques miss the multiple baselines that do not require iterative pruning but still perform exploration [1, 2, 3]. It is important to discuss how their proposed technique compares with those techniques in terms of methodology as well as experimental results.
* It would be interesting to see how the dying edge phenomenon scales with respect to the size of the network. It may be the case that for the bigger architecture model, the impact of the dying edge is less. To assess the robustness of the proposed technique for different architectures, the authors may be required to consider the higher capacity models (such as ResNet101, ViT) especially for the Cifar10 and Cifar100 datasets.

 **References:**
1. Liu, S., Chen, T., Atashgahi, Z., Chen, X., Sokar, G., Mocanu, E., Pechenizkiy, M., Wang, Z. and Mocanu, D.C., 2021. Deep ensembling with no overhead for either training or testing: The all-round blessings of dynamic sparsity. arXiv preprint arXiv:2106.14568.

2. Yin, Lu, Vlado Menkovski, Meng Fang, Tianjin Huang, Yulong Pei, and Mykola Pechenizkiy. "Superposing many tickets into one: A performance booster for sparse neural network training." In Uncertainty in Artificial Intelligence, pp. 2267-2277. PMLR, 2022.

3. Lei, B., Zhang, R., Xu, D. and Mallick, B., 2023. Calibrating the Rigged Lottery: Making All Tickets Reliable. arXiv preprint arXiv:2302.09369

**Questions:**

Experimental results stated in weaknesses section.

---

> ### Author Response · Authors · 2023-11-23
> **Response**
>
> Thank you for your insightful comments. We will answer your questions to address your concerns. Please also check the general response, in which the revision during the rebuttal period is explained.
> > One of the reasons for having increased popularity of the EP algorithm is its computational efficiency compared to iterative techniques. Specifically, without iterative pruning, we can easily get the desired subnetwork (winning ticket) in one-shot training. The proposed technique misses the key advantage of the EP algorithm as the proposed Soft iEP requires iterative pruning. Therefore, the proposed technique may be computationally expensive and may limit its applicability in crucial domains such as large language models (LLM) where the computational cost is very expensive.
>
> As you pointed out and we have originally discussed in the limitation section, the computational costs to obtain STL are one of the major drawbacks of the proposed method. On the other hand, we believe that the practical benefit of the SLT-based method is deployment-friendly since it only needs to store the binary matrix and random seed to generate weights. See the general response (1) and the revised introduction for a more detailed discussion.
>
> In addition, we updated the manuscript to clarify the concerns raised by the reviewer for the fairer comparison between the SLT-based method and the dense baseline in Fig 5. See the last paragraph of P7.
>
>
> > The proposed techniques miss the multiple baselines that do not require iterative pruning but still perform exploration [1, 2, 3]. It is important to discuss how their proposed technique compares with those techniques in terms of methodology as well as experimental results.
>
> Thank you for providing multiple references. We have carefully checked the papers, but we believe they are developed in the context of weight-base LT (multicoated lottery tickets [a]). Instead, we implemented a recently proposed extension of EP, which shares the concept of the provided literature in the sense it uses multiple (cheap) tickets to improve performance. We included the comparison with [a] in section 5 and Appendix and also verified that the proposed method can be combined with [a] to improve the performance further. See general response 2-3 as well.
>
> [a] "Multicoated Supermasks Enhance Hidden Networks", ICML2022
>
>
> > It would be interesting to see how the dying edge phenomenon scales with respect to the size of the network. It may be the case that for the bigger architecture model, the impact of the dying edge is less. To assess the robustness of the proposed technique for different architectures, the authors may be required to consider the higher capacity models (such as ResNet101, ViT) especially for the Cifar10 and Cifar100 datasets.
>
> Following the reviewer’s comment, we have included additional models, including higher capacity ResNet18 (with 2x width), ResNext, and ConvMixer. We are sorry that we have not included ViT, as ViT typically requires large-scale datasets to train. In addition, as we have discussed in the original manuscript, the performance improvement tends to be larger when we use higher capacity models. See second paragraph of section 5.

---

### Official Review · Reviewer_pwgP · 2023-10-31

**Soundness:** 2 fair
**Presentation:** 2 fair
**Contribution:** 2 fair
**Rating:** 6
**Confidence:** 3

**Summary:**

This paper empirically studies of the edges in network pruning, specifically coupling the performance of pruning with dying ratio of edges, and propose a soft edge-popup algorithm to allow selections of bottom edges, in order to improve the pruning performance.

**Strengths:**

This paper explores/defines the exploration efficacy of the pruning process from the perspective of dying edge ratio (proportion of edges have never been explored), and the empirical study and figure illustrations around this concept looks interesting and solid to me. Meanwhile, this paper specifically focus and discuss EP algorithm with important details, which I find it very helpful to understand the gist of the paper.

**Weaknesses:**

The technical or algorithmic contribution is rather limited. The soft iEP differs its hard counterpart from whether using mask, allowing the bottom edges to selected (Fig. 6 (b)). This is expected but not sure how such difference contributes to the final performance. It seems that higher exploration efficacy implies better performance, which from my point of view, has not really been given any rationales.

**Questions:**

1. It would be great to give a detailed description in formulas (Appendix should be fine) to summarize the algorithms.
2. Please give rationale and motivation of the higher exploration efficacy, the better performance?
3. For training such soft iEP, how does this converge compared to the hard one?
4. Please discuss the dying ratio and pruning performance for different randomized initializations of the network.
5. How many only visited once edges are finally retained?
6. To what degree are the final prune edges similar of soft and hard iEP edges? (maybe compare the edge graph similarity?)

---

> ### Author Response · Authors · 2023-11-23
> **Response**
>
> Thank you for your insightful comments. We will answer your questions to address your concerns. Please also check the general response, in which the revision during the rebuttal period is explained.
>
> > 1. It would be great to give a detailed description in formulas (Appendix should be fine) to summarize the algorithms.
>
> We have updated the proposed method's entire description and included the pseudocode in Appendix B.
>
> > 2. Please give rationale and motivation of the higher exploration efficacy, the better performance?
>
> Thank you for the question. I think Figure 2 in the manuscript explains the motivation of the higher exploration efficacy to provide performance improvement. Assume that we have target networks that are hidden in random networks. If the exploration efficacy is low, which means there exist many dying edges as with the standard EP, whether the algorithm can be reached at the solution strongly depends on the initial structure. In other words, low exploration efficiency means that they actually do not (or can not) use entire parameter spaces effectively, thus trapped to the suboptimal solution around the initial solution. Additional analysis of the structure difference among subnetworks obtained by Soft iEP, Hard iEP, and EP also supports this perspective (see Appendix C.3 and general response 3).
>
>
>
> > 3. For training such soft iEP, how does this converge compared to the hard one?
>
> Thank you for the question. We have included the convergence speed of hard and soft iEP. See general response 2-7 for more explanation.
>
> > 4. Please discuss the dying ratio and pruning performance for different randomized initializations of the network.
>
> We have added experiments on the dying ratio and burning performance for different randomized initializations of the network. For the dying ratio, we included the results of different weight and score initialization and how it affects the dying ratio and performance in section 3 (see general response 2-5). Besides we also tested the proposed iterative pruning with different weight initialization in Appendix C.2.4.
>
> > How many only visited once edges are finally retained?
>
> Following the reviewer's comments, we extend the analysis and compare the cumulative visit count of edges between EP and iEP. In short, EP first provides almost 10x dying edges, and Soft iEP tends to activate edges a few times. See Appendix C.3.3 for a more detailed discussion.
>
> > To what degree are the final prune edges similar of soft and hard iEP edges? (maybe compare the edge graph similarity?)
>
> Following prior works investigating the mask similarity, we have investigated mask similarity among EP, Hard iEP, and Soft iEP. See appendix C.3.2 and general response 3.

---

### Official Review · Reviewer_qaNa · 2023-11-01

**Soundness:** 3 good
**Presentation:** 3 good
**Contribution:** 3 good
**Rating:** 6
**Confidence:** 3

**Summary:**

The paper introduces a variant of the edge-popup algorithm that seeks to uncover strong lottery tickets—optimally pruned neural network architectures—by substituting the traditional rigorous pruning approach with a more lenient one. The authors delve into the concept of 'dying edges,' a seemingly-newly-introduced term coined in this paper to describe edges that are consistently overlooked during the optimization process of the milestone work edge-popup (EP), which they link to the performance of the final pruned network.

Providing a solid foundation, the paper lays out the necessary background for the newly proposed soft iterative edge popup (soft iEP). The authors build their case by discussing the limitations of the original iterative edge popup (iEP), particularly highlighting how the prevalence of dying edges can lead to inefficiencies. They draw a correlation between the quantity of dying edges and the test accuracy of the network.

Finally, the authors demonstrate the soft iEP approach effectively lowers the dying ratio and enhances performance as well; this is evidenced by experimental results on CIFAR10/100 and ImageNet using some ResNet architectures. Additionally, the paper benchmarks the proposed method against other variations of EP to provide a thorough comparison and underscore the benefits of their approach.

**Strengths:**

1. The paper is well-structured, offering sufficient background and well-supported claims with evidence to readers.
2. Soft iEP is simple yet effectively reduces dying edges, which results in performance improvement.
3. The experimental studies are well-executed offer meaningful insights.

**Weaknesses:**

1. The concept of soft pruning, also known as soft filter pruning, is not novel, having been explored in such previous works [1,2], which should be cited in this paper as it is connected to the idea of finding SLTs.

   [1] Soft Filter Pruning for Accelerating Deep Convolutional Neural Networks, IJCAI 2018 \
   [2] Operation-Aware Soft Channel Pruning using Differentiable Masks, ICML 2020

2. It would be great to see more experiments with more advanced models. For example with an advanced CNN such as ConvNeXt and the recently proposed vision transformers, the proposed method would be stronger.

3. The proposed method of soft iEP presumably has many variations for reviving edges, but the details in the manuscript are unclear to me.

**Questions:**

1. Can the authors give any (architectural) differences between the two final networks pruned by iEP and soft IEP having the same pruned weights ratio?

2.  This reviewer suggests that the paper could be strengthened by comparing the proposed method with other approaches in the domain of strong lottery tickets if such methods exist and are available.

3. This reviewer is curious about the applicability of the proposed method to pretrained models, given that hard pruning (hard-EP) has been shown to underperform with randomly initialized weights as shown in [1].

[1] Lottery Jackpots Exist in Pre-trained Models, TPAMI 2023

---

> ### Author Response · Authors · 2023-11-23
> **Response**
>
> Thank you for your insightful comments. We will answer your questions to address your concerns. Please also check the general response, in which the revision during the rebuttal period is explained.
>
> > The concept of soft pruning, also known as soft filter pruning, is not novel, having been explored in such previous works [1,2], which should be cited in this paper as it is connected to the idea of finding SLTs.
>
> Thank you for providing the missing references. We have cited them in the revised paper and discussed the connection in section 4. Besides, we have developed a new variant of soft pruning inspired by [1]. See the general response 2-6.
>
>
> > It would be great to see more experiments with more advanced models. For example with an advanced CNN such as ConvNeXt and the recently proposed vision transformers, the proposed method would be stronger.
>
> We have included other backbone networks, including (ResNet18x2, RexNext, and ConvMixer). Due to resource limitations (lack of A100 GPU), we have only conducted experiments on the CIFAR10 and CIFAR100, but we believe the results are enough to show their generality. See general response 2-2.
>
> > The proposed method of soft iEP presumably has many variations for reviving edges, but the details in the manuscript are unclear to me.
>
> Sorry for the confusion. Since soft iEP is only valid for the LR and FT setup (if we use SR, it degenerates to the standard EP), we used the LR option throughout the paper. We have clarified on this point in the sanction 4.
>
> > Can the authors give any (architectural) differences between the two final networks pruned by iEP and soft IEP having the same pruned weights ratio?
>
> Thank you for the insightful comments. Inspired by the reviewer’s comment, we have conducted additional analysis on how different the obtained subnetworks are. Please see the general response 3.
>
>
> > This reviewer suggests that the paper could be strengthened by comparing the proposed method with other approaches in the domain of strong lottery tickets if such methods exist and are available.
>
> Following the reviewer’s comment, we included a new baseline. Please see the general response 2-3.
>
>
> > This reviewer is curious about the applicability of the proposed method to pretrained models, given that hard pruning (hard-EP) has been shown to underperform with randomly initialized weights as shown in [1].
>
> Thank you for the interesting question. We have tested the applicability of soft iEP to the pre-trained weights. See general response 2-4.

---

### Official Review · Reviewer_ce4d · 2023-11-04

**Soundness:** 3 good
**Presentation:** 3 good
**Contribution:** 3 good
**Rating:** 5
**Confidence:** 3

**Summary:**

This paper studies the Strong Lottery Tickets Hypothesis (SLTH), which suggests that randomly initialized dense network itself contains sparse subnetworks that can achieve comparable performance with dense network. Specifically, this paper first explores the Edge Popup (EP) algorithm of SLTH that learns the sparse mask with a popup score, rather than update the weights, and empirically finds that EP results in a high ratio of dying edges, which is the edge that is never selected until the termination of the algorithm. As a result, the performance of the SLTH is hindered. To address this issue, this paper proposes Soft Iterative EP (Soft IEP). Soft IEP is the first attempt that applies the common iterative pruning method from LTH to SLTH. Besides, Soft IEP also suggests soft pruning, where are edges can be selected at the end of the EP. Experiments results on CIFAR and ImageNet with different architectures demonstrate the effectiveness of the proposed Soft IEP.

**Strengths:**

- The topic about SLTH is interesting, which only learns the mask (structure) of the subnetwork, rather than the weights. Besides, the empirical analysis of EP from the dying ratio perspective is interesting, which has never been explored before. Moreover, this paper also present many evidence to support the claim about dying ratio.

- The proposed solution with iterative pruning and soft edge is simple yet effective and well-motivated.

- This paper is well-written and easy to follow.

**Weaknesses:**

- It’s better if this paper could discuss more benefits of SLTH. What is the major benefits of only learning mask, rather than weights? What is the current performance gap between learning mask and learning weights.  For example, based on Figure 5, we can see that a ResNet-50 with 8 millions number of parameters only achieve a performance around 70%, while at this 66% (1 - 8 / 23) sparsity level, many previous LTH with unstructured pruning  have show that they can still almost match the performance of the dense net (~76%). Thus, there is still a huge performance gap for SLTH-based methods. I would suggest the authors highlight the practical benefit of SLTH, and also show the performance gap with weight-based LTH.

- The novelty of applying iterative pruning to the SLTH is kind of limited. Although this paper claims that iterative pruning has not been evaluated in SLTH, the proposed idea is not exciting considering that iterative pruning has been massively explored in the LTH community.


- The investigation of the relationship between dying ratio and performance may be unclear. This paper explores different hyperparameters to optimize EP, including batch size, learning rate, etc. However, it’s unclear whether these hyper parameters themselves will hinders the performance. For example, it’s common that different optimizer will result in different performance when you learn the weights. Thus, the causal relationship between them (“”high dying ratio hinders the performance) cannot be verified. Besides, it’s unclear why change these hyperparameters can result in different dying ratio. It’s better for this paper to illustrate their relationship.


- It would be advantageous to include mathematical formulations for the proposed soft edge. The current draft employs plain language to convey the concept, ensuring ease of comprehension but lacking the necessary technical rigor.

**Questions:**

Please address the above issues.

---

> ### Author Response · Authors · 2023-11-23
> **Response**
>
> Thank you for your insightful comments. We will answer your questions to address your concerns. Please also check the general response, in which the revision during the rebuttal period is explained.
>
> >I would suggest the authors highlight the practical benefit of SLTH, and also show the performance gap with weight-based LTH.
>
> Following the reviewer's comments, we have updated the introduction to discuss the practical benefit of the SLTH. In short, it is more robust to the quantization and, moreover, it can easily recover weight matrix only via random seed. Thus, only need to store binary matrix during the deployment, providing practical merits on edge device scenario. We have clarified this point mainly in the revised introduction.
>
> On the other hand, the SLT-based method still usually underperforms the method involving weight updates like dynamic sparse training, the standard lottery tickets, and its variants. We have clarified this point as well in the experiment parts (section 5). Besides, we have conducted additional experiments to clarify this point in Appendix C.2.1; see the general response 2-1.
>
>
> > The novelty of applying iterative pruning to the SLTH is kind of limited. Although this paper claims that iterative pruning has not been evaluated in SLTH, the proposed idea is not exciting, considering that iterative pruning has been massively explored in the LTH community.
>
> I agree that iterative pruning itself is widely used in the LTH community; however, as we have discussed in the introduction, the application to SLTH is limited. In this sense, our paper can be posed as the first empirical analysis of the effectiveness of iterative pruning on SLTH, using a variety of datasets (CIFAR10, CIFAR100, and ImageNet), architecture (CNN, ResNet, ResNext, and Conv Mixer), retraining procedure (hard pruning, soft pruning, and various rewinding procedure). In addition, our main novelty is also posed in the analysis of the pitfall of the SLTH-based method (the dying edge problem).
>
> > Thus, the causal relationship between them ("" high dying ratio hinders the performance) cannot be verified.
>
> I'm sorry for the confusion. I agree that this part is misleading and overclaimed; thus, I revised and moved it to the appendix. We agree that it is difficult to investigate the causal relationship and thus do not intend to claim that.
>
> > Besides, it's unclear why change these hyperparameters can result in different dying ratio. It's better for this paper to illustrate their relationship.
>
> Thank you for the comments. We have included the discussion on this point in appendix A.2. While it might be a bit obvious, learning rate and batch size affect the dying ratio since the larger learning rate (or equivalently smaller batch size) possibly swaps more edges. Similarly, the weight decay reduces the magnitude of the score, making it easier to swap edges with smaller gradients.
>
> > It would be advantageous to include mathematical formulations for the proposed soft edge. The current draft employs plain language to convey the concept, ensuring ease of comprehension but lacking the necessary technical rigor.
>
> Thank you for the comment. As we mentioned in the general comment, we have tried to improve the technical rigor of the proposed method.

---

### Author Response · Authors · 2023-11-23
**General response**

We thank all the reviewers for carefully reading our paper and giving insightful comments. Based on the reviewers' feedback, we have updated the paper to address their concerns. The updated part is highlighted with colored text.


**(Major Revision 1) Clarify the practical benefit and drawbacks of the SLT-based methods compared to dense training, including the standard LT** (ce4d,tFae)

We have clarified the practical benefits and drawbacks of the SLT-based method. In short, it is more robust to the quantization and, moreover, it can easily recover weight matrix only via random seed. Thus, we only need to store the binary matrix during the deployment, providing practical merits in resource-limited scenarios (including edge devices). We have clarified this point mainly in the revised introduction. This unique property of SLT, including the interesting fact that it does not require any weight training, motivates us to develop an algorithm to find better SLT. Thus, comparison among SLTH baseline is enough to show the merits of the proposed method.

On the other hand, the SLT-based method still usually underperforms the method involving weight updates like dynamic sparse training, the standard lottery tickets, and its variants. We have clarified this point in the experiment parts (section 5). Besides, we have conducted additional experiments to clarify this point, as discussed later in the thread.


**(Major Revision 2) We have conducted additional experiments.**

Following reviewer's requests, we have conducted additional experiments from various aspects.


**(2-1) Comparison with dense lottery tickets. (tFae)**

We have added the simple dense LT baseline in appendix C.2.1. Our method shrinks the gap between dense LT and SLT. However, we confirmed that there exists a performance gap, especially when using a high sparsity region. We also confirm that the performance gap can be further shrunk by combining the recently proposed extension of SLTH (see 2-3).

**(2-2) Comparison of different architecture. (qaNa,He2e)**

We have compared EP and Soft iEP on a different architecture (ResNet18x2, ResNext, and ConvMixer) in Figure 4 and Appendix C.2.3. We have confirmed that Soft iEP stably increases the performance regardless of the choice of the architecture.


**(2-3) Comparison with advanced SLTH baseline (qaNa,He2e)**

We have added a new baseline recently proposed in the context of SLT [1]. We confirmed that our proposed method outperforms [1] and can be combined to find better SLT, especially in the high-sparsity region.

[1] "Multicoated Supermasks Enhance Hidden Networks", ICML2022

**(2-4) Applicability of the proposed method to prune the pre-trained weights (qaNa)**

We apply our method to prue the pre-trained weights. While we found that the Soft iEP generally increases the performance compared to EP, the subnet from pre-trained weights provides on-par performance with a subnet from random weights. We have included the discussion in Appendix C.2.4.


**(2-5) An effect of weight and score initialization on the dying ratio (pwgP)**

We revised section 3 and included the effect of the initialization on the dying edge problems. In short, results suggest that the dying edge problem exists regardless of the choice of standard initialization. At the same time, we have revised the correlation analysis between accuracy and the dying ratio, which might be misleading and overclaimed, as the reviewer page suggested. Due to the space limitation, we moved it to the appendix. In addition, we added a discussion on why some hyperparameters naturally reduce the dying ratio in Appendix A.2.


**(2-6) Comparison with different soft pruning (tFae,qaNa)**

We agree that the proposed method shares a similar split with soft filter pruning, as reviewed by reviewer qaNa, while it is designed for weight training and thus can not be directly applied to our setting. We have cited the relevant paper and designed another variant of soft pruning inspired by the existing literature (appendix C.2.1).

**(2-7) Convergence speed of Hard and Soft iEP (pwgP)**

We compare the convergence speed of Hard and Soft iEP (appendix C.2.5). As a reader may imagine, Hard iEP generally performs well at the beginning of the training, maybe due to its optimized parameters, but Soft iEP gradually increases performance and outperforms at the end. In other words, Hard iEP speeds up the optimization at the risk of a suboptimal solution. Combining hard and soft iEP is an interesting future topic.

---

> ### Author Response · Authors · 2023-11-23
> **cont.**
>
> **(Major Revision 3) We have further analyze the structural difference of the obtained subnetworks** (qaNa, pwgP)
>
> We agree that understanding the structure difference is an interesting analysis and might provide insight into designing a better algorithm. In Appendix C.3, we conducted a set of analyses to clarify the difference.
>
> In Appendix C.3.1, we first investigated how the trained networks are different among EP, Hard iEP, and Soft iEP. As a result, EP and Soft iEP provide almost completely different architecture (almost chance level). Hard iEP also provides substantially different subnetworks from Soft iEP, and the difference gets larger as we repeat the pruning procedure, as a prior study in LTH also reported [2].
>
> Secondly, to analyze the in-depth difference among structures, we count the dying neuron that exists in the second last layer (see appendix C.3.2). Specifically, we count the number of neurons that have no input-side connection and neurons that have no output-side connection. As a result, we found that EP and Hard iEP provide a lot of neurons that have no output-side connection but still have an input-side connection. Since the neuron does not affect the prediction, the edges connected to such a neuron must be useless. On the contrary, in the case of Soft iEP, most neuron with no output-side connection also does not contain any input-side connection (92/94 and 326/331), which means that the Soft iEP assign the scores to potentially more useful edges.
>
>
> [2] "Bespoke vs. Prêt-à-Porter Lottery Tickets: Exploiting Mask Similarity for Trainable Sub-Network Finding"
>
>
>
> **(Major Revision 4) We have updated the explanation to improve the rigor of the explanation** (ce4d, pwgP)
>
> Some reviewers request improving the technical rigor. We have proof-read the paper and tried to improve the rigor throughout the paper. Table 3 also summarizes the main difference between the baselines. In addition, we added a pseudo-code in Appendix B.

---

### Meta-Review · Area_Chair_xtcP · 2023-12-18

**Metareview:**

This paper proposes an improved version of the Edge-popup (EP) algorithm for finding so-called strong lottery tickets (SLT) - masks that, when applied to a randomly initialized network, result in good performance on a particular task. This paper looks like an incremental improvement. Reviewers were lukewarm, with concerns about the computational cost, comparisons to the rest of the pruning literature, and quality of the empirical work. I add one more concern: this isn't an especially compelling problem in general, given that it's a subproblem of a subproblem and this is an incremental advance. There simply isn't much room for significance. The paper is borderline, but - alongside the reviewers - I lean toward rejection.

**Justification For Why Not Higher Score:**

The paper is okay, but the improvement is incremental and the problem just isn't that important. It's a made-up academic problem that is a subproblem of another made-up academic problem (lottery tickets).

**Justification For Why Not Lower Score:**

N/A

---

### Decision · Program_Chairs · 2024-01-16

Reject